# Scaling Retrieval-Based Language Models with a Trillion-Token Datastore

**Rulin Shao**[1]  **Jacqueline He**[1]  **Akari Asai**[1]  **Weijia Shi**[1]

**Tim Dettmers**[1]  **Sewon Min**[1]  **Luke Zettlemoyer**[1]  **Pang Wei Koh**[1,2]

[1]University of Washington  [2]Allen Institute for AI

{rulins,jyyh,akari,swj0419,dettmers,sewon,lsz,pangwei}
@cs.washington.edu

## Abstract

Scaling laws with respect to the amount of training data and the number of parameters allow us to predict the cost-benefit trade-offs of pretraining language models (LMs) in different configurations. In this paper, we consider another dimension of scaling: the amount of data available at *inference* time. Specifically, we find that increasing the size of the datastore used by a retrieval-based LM monotonically improves language modeling and several downstream tasks without obvious saturation, such that a smaller model augmented with a large datastore outperforms a larger LM-only model on knowledge-intensive tasks. By plotting compute-optimal scaling curves with varied datastore, model, and pretraining data sizes, we show that using larger datastores can significantly improve model performance for the same training compute budget. We carry out our study by constructing a 1.4 trillion-token datastore named MASSIVEDS, which is the largest and the most diverse open-sourced datastore for retrieval-based LMs to date, and designing an efficient pipeline for studying datastore scaling in a computationally accessible manner. Finally, we analyze the effect of improving the retriever, datastore quality filtering, and other design choices on our observed scaling trends. Overall, our results show that datastore size should be considered as an integral part of LM efficiency and performance trade-offs. To facilitate future research, we open-source our datastore and code at https://github.com/RulinShao/retrieval-scaling.

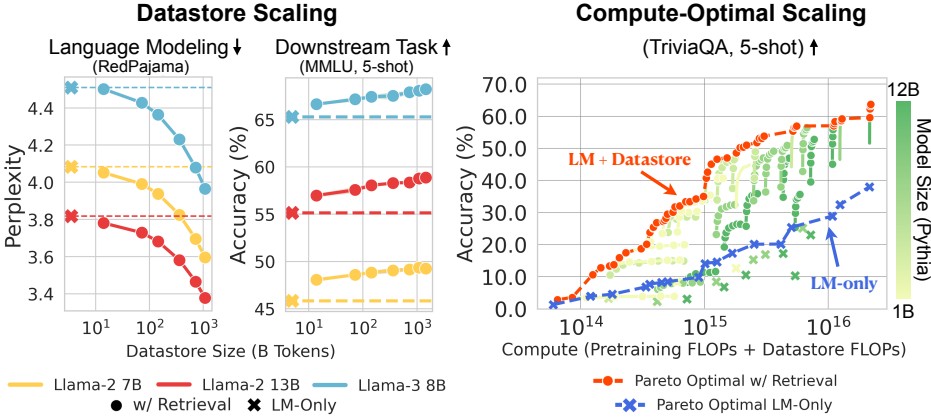

Figure 1: **Datastore scaling improves language modeling and downstream task performance.** *Left:* Datastore scaling performance on language modeling and a downstream task (MMLU) with LLAMA-2 and LLAMA-3 models. *Right:* Compute-optimal scaling of retrieval-based language models vs. LM-only models with PYTHIA models. By considering the size of the datastore as an additional dimension of scaling, we can improve model performance at lower training cost.

38th Conference on Neural Information Processing Systems (NeurIPS 2024).

# 1  Introduction

The scaling of large language models (LMs) has driven tremendous performance gains across a variety of tasks (Brown et al., 2020; Kaplan et al., 2020; Muennighoff et al., 2023). Current scaling laws are primarily a function of the size of the pretraining data and the number of parameters (Hoffmann et al., 2022; Muennighoff et al., 2023; Gadre et al., 2024). In this paper, we consider another dimension of scaling: the amount of data in a datastore used at *inference* time by retrieval-based LMs, which can directly retrieve information from the datastore to use in context when generating output (Karpukhin et al., 2020; Guu et al., 2020; Izacard & Grave, 2020; Asai et al., 2024b).

Retrieval-based LMs have a range of benefits such as improved factuality (Mallen et al., 2023), effective domain adaptation (Khandelwal et al., 2020), credit attribution (Gao et al., 2023), and parametric efficiency (Min et al., 2023b). However, most prior work in retrieval-based LMs use datastores constructed from a single data source (Karpukhin et al., 2020), such as Wikipedia, with sizes on the order of a few billion tokens. While there has been some work on larger datastores (Table 1), with the largest being RETRO (Borgeaud et al., 2022; Wang et al., 2024) in the trillion-token range, these studies use proprietary datastores and custom architectures with a limited evaluation suite. As such, it remains unknown how datastore scaling helps the currently dominant retrieval-in-context approaches on a broad categories of tasks.

We first construct MASSIVEDS, a massively multi-domain datastore comprising 1.4 trillion tokens of both general web data and domain specific data (§3.1) that serves as the cornerstone for our scaling study. A key challenge in studying datastore scaling is the computational cost introduced by building datastores with all possible combinations of factors such as the datastore scale, data composition, random seed for subsampling, and different data preprocessing methods. To make our study accessible, we design an efficient datastore construction pipeline that reduces the compute needed by an order of magnitude while being equivalent to the standard pipeline (§3.2).

Using the proposed pipeline, we systematically evaluate the effects of scaling MASSIVEDS on retrieval-based LMs with varying numbers of parameters and pretraining tokens (§4). Beyond upstream language modeling, we also consider a suite of diverse downstream tasks, including general-knowledge question answering (QA), domain-specialized QA, and reasoning tasks. We find that, first, datastore scaling consistently improves both language modeling and some downstream tasks in a task-dependent manner (Figure 1 *Left*), much like the widely observed data and parameter scaling trends. In fact, on knowledge-intensive tasks, a small retrieval-based LM can outperform its larger LM-only counterparts. Second, since indexing a datastore is cheaper than training on the same amount of data, retrieval-based LMs enable better compute-optimal scaling trends, where they achieve superior performance than LM-only models at the same training cost (Figure 1 *Right*).

Through our analyses (§5), we show that retrieval-based LMs are capable of automatically retrieving documents that are in-domain to the query, which allows them to reap the benefits of larger, broader datastores. In addition, data quality filters and improved retrieval methods can further enhance our observed scaling trends.

Overall, our results show that datastore size should be considered as an integral part of LM efficiency and performance trade-offs. To spur future research, we open-source MASSIVEDS (including the raw passages, the embedding, and the index) and all code (including our evaluation suite and pipeline for efficiently studying datastore scaling) at `https://github.com/RulinShao/retrieval-scaling`.

# 2  Related Work

**Retrieval-based LMs.** Unlike parametric LMs that only use data during training, retrieval-based LMs can access data through a datastore during inference (see Asai et al. (2024b) for a review). We focus on retrieve-in-context language models (RIC-LMs), which retrieves a small set of documents from the datastore and feeds a concatenation of them as an input to the LM (Ram et al., 2023; Shi et al., 2023). The RIC-LM approach is simple and allows the use of off-the-shelf retrievers and LMs, even with only black-box access.

**Scaling the retrieval datastore.** Prior work on retrieval-based LMs often focused on specific aspects of LMs such as factuality and attribution. In addition, they typically use limited-size, single-domain

Table 1: **Comparison with prior work, ordered by datastore size.** '# Tokens' indicates the number of tokens in the datastore using the LLAMA2 tokenizer (Touvron et al., 2023). The asterisk (*) denotes that the datastore is not evaluated on downstream tasks. MASSIVEDS is the largest open-sourced datastore and covers a broad spectrum of domains.

| Reference | # Tokens | Data Sources | Open sourced |
|---|---|---|---|
| ATLAS (Izacard et al., 2023) | <5B | Wikipedia | ✗ |
| REALM (Guu et al., 2020) | <5B | Wikipedia | ✗ |
| RALM (Ram et al., 2023) | <5B | Wikipedia | ✓ |
| SELF-RAG (Asai et al., 2024a) | <5B | Wikipedia | ✓ |
| REPLUG (Shi et al., 2023) | 47B | The Pile | ✓ |
| RA-DIT (Lin et al., 2024) | 79B | Wikipedia, CommonCrawl | ✗ |
| SPHERE (Piktus et al., 2022) | 90B | CCNet | ✓ |
| RETRO++ (Wang et al., 2024) | 330B* | The Pile, CommonCrawl, RealNews, CC-Stories | ✗ |
| INSTRUCTRETRO (Wang et al., 2024) | 1.2T* | Wikipedia, CommonCrawl, RealNews, CC-Stories, Books | ✗ |
| RETRO (Borgeaud et al., 2022) | 1.7T* | MassiveText (Rae et al., 2022) | ✗ |
| MASSIVEDS (Ours) | **1.4T** | 8 domains, listed in Table 2 | ✓ |

datastores such as Wikipedia, which is on the order of a few billion tokens (Table 1). Scaling the datastore remains relatively underexplored, with two notable exceptions. First, Borgeaud et al. (2022) proposed a new RETRO transformer architecture for retrieval-based LMs and built a 1.75 trillion token datastore sourced from the proprietary data introduced in Rae et al. (2022). However, RETRO and its follow-up work, RETRO++ (Wang et al., 2023) and INSTRUCTRETRO (Wang et al., 2024), use this trillion-token datastore solely for language modeling evaluation, while using a small task-specific Wikipedia datastore for downstream task evaluation. Since RETRO-based datastores are not fully open-sourced, it is challenging to replicate work based on RETRO to assess the effectiveness of datastore scaling. Second, Piktus et al. (2022) proposed SPHERE, an open-sourced 90 billion-token datastore curated from CCNet (Wenzek et al., 2020). However, their evaluation on downstream tasks such as KILT (Petroni et al., 2021) suggests that SPHERE does not always outperform a small, in-domain datastore like Wikipedia.

In contrast, we present the first study on the downstream performance of trillion-token scale datastores, including an analysis of compute-optimal scaling trends using retrieval-based LMs with different sizes of datastores, models, and pretraining corpora. Our work is fully open-source and can be replicated on a limited computational budget, enabling research on trillion-token datastores to be more broadly accessible.

## 3 MASSIVEDS and our Datastore Scaling Pipeline

### 3.1 MASSIVEDS: A Trillion-Token Datastore With a Diverse Domain Composition

MASSIVEDS encompasses both *domain-specific* data and *general web* data (Table 2). Domain-specific data comes from specialized sources, and tends to be smaller but higher in quality. We cull from a mix of data-rich domains: **books** which span a variety of genres (Computer, 2023); open-access **scientific papers** (Lo et al., 2020; Soldaini & Lo, 2023; Computer, 2023); **encyclopedic articles** (Karpukhin et al., 2020; Computer, 2023); **community questions and answers** from StackExchange (Computer, 2023); **code** from GitHub (Computer, 2023); **mathematical webpages** (Paster et al., 2023) and **mathematical language** (Welleck et al., 2021); **biomedical articles** (of Medicine, 2023). On the other hand, general web data

Table 2: The domain-wise data composition of MASSIVEDS. RPJ denotes REDPAJAMA V1 (Computer, 2023), CC denotes Common Crawl, Wiki denotes Wikipedia.

| Domain | Datasets | Size (B) |
|---|---|---|
| BOOKS | RPJ Books | 26.3 |
| STEM | peS2o, RPJ ArXiv | 97.7 |
| ENCYCLOPEDIA | DPR 2018 Wiki, RPJ 2022 Wiki | 31.9 |
| FORUM (Q&A) | RPJ StackExchange | 20.2 |
| CODE | RPJ Github | 52.8 |
| MATH | OpenWebMath, NaturalProofs | 14.1 |
| BIOMEDICAL | PubMed | 6.5 |
| GENERAL WEB | RPJ CC (2019–2023), RPJ C4 | 1191.7 |
| **Total** | | 1441.2 |

is sourced from Common Crawl snapshots at five time periods (07/2019, 05/2020, 04/2021, 05/2022, 06/2023) and C4 (Raffel et al., 2020).

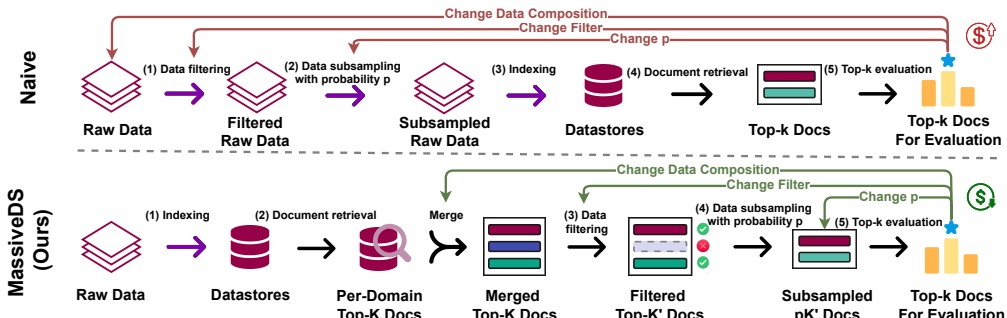

Figure 2: **Comparison between the MASSIVEDS pipeline and a naive pipeline for studying datastore scaling (§3.2).** The green and red arrows indicate repeated operations. In the naive pipeline, these operations are more expensive because they require repeating expensive steps such as rebuilding the datastore index. In the MASSIVEDS pipeline, these operations are cheaper because they repeat fewer steps and are only run on the retrieved top-K documents instead of the full datastore. Datastore-level operations are represented by purple arrows, while document-level operations, repeated for every query, are represented by black arrows.

## 3.2 Studying Datastore Scaling with the MASSIVEDS Pipeline

Studying datastore scaling requires constructing datastores of varying sizes and varying compositions from the raw text corpus. This involves the following operations: **data filtering**, including deduplication, decontamination, and quality filters (Soldaini et al., 2024); **data subsampling**, which randomly subselects a $p$-fraction of the text corpus to achieve the specified size; **indexing**, which embeds the data using an auxiliary model and builds a searchable index; **document retrieval**, which uses the index to find the top-$k$ relevant documents for each test query[1]; and **top-$k$ evaluation**, which uses the top-$k$ documents per test query to augment the retrieval-based model. A naive approach is to run these operations in the aforementioned order for each datastore, and build separate datastores for all combinations of subsampled datastore sizes, random seeds, and other experimental variations. However, this naive approach is prohibitively expensive[2] at the trillion-token datastore scale because it repeats expensive operations, as shown in Figure 2 (top).

To make the datastore scaling study feasible, we develop the MASSIVEDS pipeline (Figure 2 bottom). The key idea is to reorder the above operations such that the most expensive ones—indexing and retrieval—are run only once at the start and then shared across all subsequent datastore variants. Other operations with many variants—such as subsampling, deduplication, and decontamination—are performed as late as possible to minimize repeating subsequent steps. To enable this, we first retrieve a relatively large number ($K \gg k$) of documents for each query and then apply the subsequent operations to these sets of retrieved documents, rather than the entire datastore. Altogether, this pipeline reduces compute, I/O, and storage requirements by more than an order of magnitude, enabling us to conduct the datastore scaling study on a modest compute budget. In Appendix A.5, we show that the results from the MASSIVEDS pipeline are equivalent to the results from the naive pipeline with high probability, where the randomness comes from the subsampling procedure. We provide more details on the steps in the MASSIVEDS pipeline in Appendix A and detailed configuration in Appendix B.1.

**Note on the number of tokens in the datastore.** In our figures, we plot the datastore size (on the x-axis) by multiplying the total number of tokens in the raw data pool by the subsampling fraction $p$. A more accurate representation would be the number of tokens in the filtered data pool; however, we do not know the size of the filtered data pool as we apply data filtering on the retrieved documents instead of the raw data for computational efficiency. As a result, the number of tokens we plot on our x-axis is proportionally larger, i.e., if a $p_f$ fraction of the data is filtered out ($0 < p_f \leq 1$), then the

---

[1]Reranking is optionally applied, which reranks the retrieved documents based on a more effective but usually more expensive heuristic (Nogueira & Cho, 2019; Sachan et al., 2022).

[2]The operations involved in the naive pipeline are expensive in terms of FLOPs, I/O operations, and storage due to repetitive index construction and large-scale retrieval.

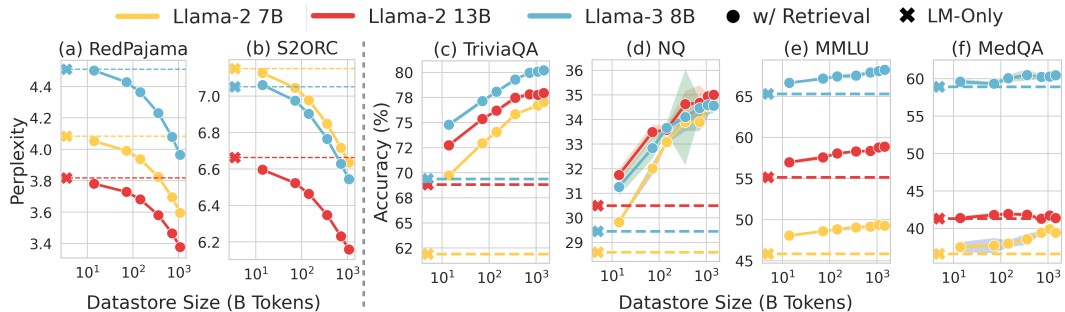

Figure 3: **Scaling performance on upstream and downstream tasks with MASSIVEDS, in comparison with LM-only performance.** *Left:* Perplexity (PPL) scaling performance on REDPAJAMA (multi-domain pretraining corpus) and S2ORC (scientific papers). *Right:* Downstream scaling performance on TriviaQA (TQA), Natural Questions (NQ), MMLU, and MedQA.

actual number of tokens should also be multiplied by $p_f$. Since we plot datastore size on a log axis, this corresponds to a constant shift and does not change the scaling trends.

## 4 Datastore Scaling with Retrieval-Based Language Models

### 4.1 Experimental Setup

**Model details.** Following prior work (Izacard et al., 2023; Liu et al., 2023; Ram et al., 2023; Shi et al., 2024; Xu et al., 2023; Asai et al., 2024a), we use CONTRIEVER-MSMARCO (Izacard et al., 2022), which represents every document in the datastore as a dense vector, as our retriever. We ablate the choice of retriever in Appendix E.1; we found that CONTRIEVER-MSMARCO performs on par with, or even better than, more recent larger retrievers. We augment input examples with retrieved documents at the granularity of 256-word chunks. We study datastore scaling performance with the LLAMA-2, LLAMA-3 (Touvron et al., 2023), PYTHIA (Biderman et al., 2023), and OLMO (Groeneveld et al., 2024) model families.

**Evaluation.** We consider both language modeling and downstream tasks for evaluation. We evaluate language modeling perplexity on data from two domains: (1) general web data sampled from REDPAJAMA (Computer, 2023); (2) scientific paper data sampled from S2ORC (Lo et al., 2020). For downstream tasks, our evaluation encompasses general-knowledge, medical, math, and science domains including the following tasks. TriviaQA (**TQA**; Joshi et al. 2017) comprises trivia questions with answers sourced from Wikipedia and the web. Natural Questions (**NQ**; Kwiatkowski et al. 2019; Lee et al. 2019) comprises search engine queries and human-annotated answers from Wikipedia. Massive Multitask Language Understanding (**MMLU**; Hendrycks et al. 2021) comprises general-purpose, multi-task reasoning questions. **MedQA** (Jin et al., 2020) comprises medical multiple-choice questions sourced from professional medical exams.

**Implementation details.** For evaluation with retrieval, we concatenate the top $k = 3$ documents in reverse order, so that higher-ranked documents are positioned closer to the query. For downstream tasks, we evaluate models via 5-shot prompting, and we prepend the retrieved documents before the few-shot examples, followed by the question. We do not apply reranking for our main experiments in Section 4; we study the effect of rerankers in Section 5.2. More details, including decontamination measures, are in Appendix B.

### 4.2 Datastore Scaling Results on Language Modeling and Downstream Tasks

**Finding 1: Datastore scaling significantly helps language modeling.** Figures 3(a) and (b) show perplexity curves as a function of datastore size on general web and scientific papers, respectively. Retrieval is strictly beneficial for language modeling: the LM-only baselines (denoted by dashed lines) show the highest perplexity across all models and evaluation datasets. Scaling up the datastore reduces perplexity without signs of saturation, suggesting that further scaling is likely to yield additional

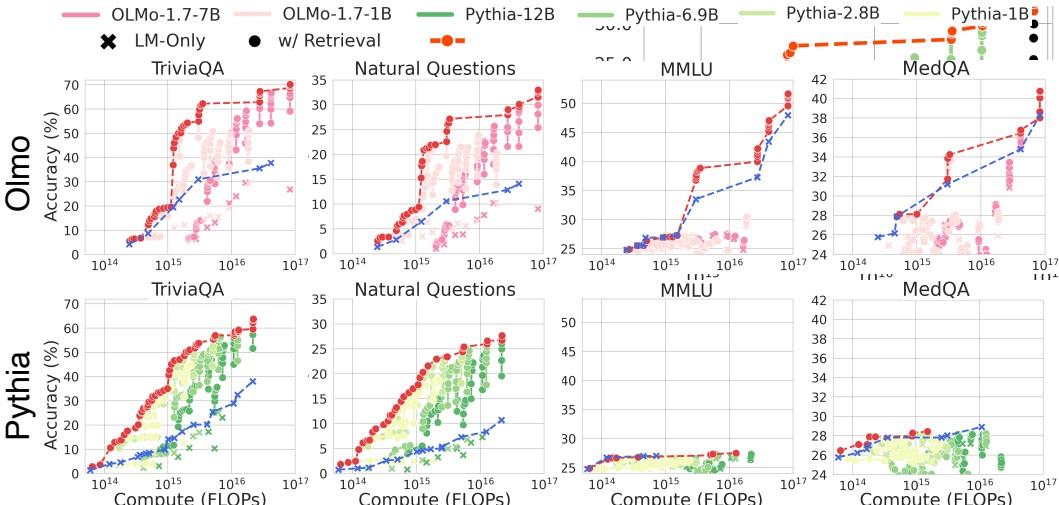

Figure 4: **Compute-optimal scaling curves for retrieval-based and LM-only models of varying datastore sizes, model sizes, and pretraining corpus sizes (detailed setup in §B.4).** Darker green or pink indicate larger model sizes for PYTHIA and OLMO respectively; crossmarks in matching colors represent the same model size trained with varying numbers of tokens; each crossmark corresponds to a datastore scaling curve of lined dots similar to the ones in Figure 3. The Pareto-optimal points are highlighted in red for retrieval-based LMs and blue for LM-only. Within a fixed computational budget (represented on the x-axis), retrieval-based LMs achieve superior performance, which remains unsaturated along the datastore scaling dimension. Pythia models do not exhibit meaningful scaling curves on MMLU and MedQA that require advanced reasoning abilities.

improvements. Further, datastore scaling enables small models to outperform their larger LM-only counterparts: when retrieving from MASSIVEDS at the largest scale, LLAMA-2 7B outperforms the LM-only performance of its larger LLAMA-2-13B counterpart. Interestingly, we find LLAMA-3 8B underperforms LLAMA-2 7B on RedPajama. This aligns with the observations in Xiao et al. (2023) and we discuss potential reasons in Appendix D.

**Finding 2: Datastore scaling improves performance on several downstream tasks, and the degree of improvement is task-dependent.**    Figure 3(c)–(f) show the performance on four downstream tasks as a function of datastore size. Datastore scaling brings major improvements to knowledge-intensive question answering tasks such as NQ and TQA, where retrieval-based LMs significantly outperform LM-only baselines across all scales, and performance monotonically increases with datastore scale. For instance, a LLAMA-2 7B model that retrieves from fewer than 100B tokens can outperform both its 13B LM-only counterpart and the more capable LM-only LLAMA-3 8B on TQA and NQ, indicating the effectiveness of storing knowledge in the datastore.

On MMLU, a multi-subject, reasoning-heavy benchmark, datastore scaling monotonically improves performance across all model scales. Results are more mixed for MedQA, where only the weaker LLAMA-2 7B benefits more from datastore scaling. For both tasks, datastore scaling does not help the smaller model do better than the larger model. We suspect that this is due to task difficulty and the lack of in-domain data sources: both MMLU and MedQA are more oriented toward reasoning rather than pure factual recall, which poses bigger challenges for both the retriever and the LM. Additionally, MASSIVEDS only contains a small subset of web data and medical papers which may not cover all necessary information to answer these questions. We defer to future work to explore better data sources for these tasks.

## 4.3   Compute-Optimal Scaling with Retrieval-Based Language Models

Next, we study performance as a function of total training-time compute and show that retrieval-based LMs achieve superior compute-optimal performance compared to LM-only models.

**Use of intermediate checkpoints.** We use the intermediate checkpoints of PYTHIA and OLMO as an approximation of models trained on different numbers of tokens, as detailed in Appendix B.4. These intermediate checkpoints share the same learning rate scheduler, with a fixed maximum number of training steps that equals or exceeds the number of steps they have been actually trained for, and therefore the performance of these intermediate checkpoints (with or without retrieval) might be lower than otherwise attainable with the same amount of compute. However, pretraining LMs from scratch for all combinations of model sizes and numbers of pretraining tokens is prohibitively expensive for an academic budget.

**FLOPs calculation.** We detail the FLOPs computation for datastore construction and pretraining in Appendix B.4. Datastore construction is much cheaper than pretraining because it only requires one forward pass on all tokens in the datastore with a small retriever (177M parameters in our setup), while pretraining requires a forward pass and a backward pass on pretraining tokens with an LM that can be much larger than the retriever. As we use a flat index, no additional operations are required at the indexing step, so the number of FLOPs for datastore construction equals the number of FLOPs for embedding. We note that other types of indexing, e.g., inverted file indexing (IVFADC) (Jégou et al., 2011), may require additional FLOPs during construction and fewer FLOPs at inference. We first focus on training-time compute and discuss inference cost at the end of the section.

We show the scaling curves against computational cost of retrieval-based LMs and LM-only performance on downstream tasks in Figure 4. The Pareto-optimal points for retrieval-based and LM-only settings are highlighted in red and blue, respectively.

**Finding 3: Retrieval-based LMs outperform LM-only models for the same compute budget.** With the same training-time compute, retrieval-based LMs achieves superior performance than LM-only models, indicating offloading FLOPs from pretraining to datastore construction can result in better performance. Therefore, we conjecture that storing factual knowledge in a datastore is more computationally efficient than memorizing factual knowledge in model parameters at training time. We note this claim assumes the LM has enough capacity to reason with the retrieved knowledge. Otherwise, an LM may fail to utilize the retrieved knowledge, which we further discuss in Finding 5.

**Finding 4: Even weak language models can benefit significantly from retrieval on knowledge-intensive tasks that measure factual recall.** Surprisingly, we find that retrieval-based PYTHIA models (trained on up to 300B tokens) and OLMO-1.7 models (trained on up to 2T tokens[3]) have a similar compute-optimal scaling trajectory on TriviaQA and NQ (left columns in Figure 4), despite PYTHIA being trained on less and lower-quality data. Both TriviaQA and NQ evaluate factual recall without complex reasoning. When the right information is provided in context using retrieval, the LM only needs to extract the answer; therefore, these results suggest that the ability to extract factual knowledge for simple factual question answering is obtained early in training.

**Finding 5: Retrieval shows benefits for reasoning-intensive tasks with capable OLMO models, but it does not help when the language model is not sufficiently advanced such as PYTHIA.** As shown on the right side of Figure 4, datastore scaling gives marginal benefits on MMLU and MedQA for PYTHIA models where the performance stays around random even at the 12B model size. However, OLMO, which is trained on more and better data, consistently benefits from retrieval on both tasks. We thus conjecture that training on higher-quality data, as OLMO applied in pretraining, could help the model benefit more from retrieved documents for reasoning-heavy tasks. Beyond reasoning ability, access to the right data sources for the datastore might be critical. For example, we observe fewer benefits from retrieval on MMLU and MedQA in comparison with TriviaQA and NQ. This may indicate that MMLU and MedQA need more specific data, such as relevant textbooks for MMLU and biomedical literature for MedQA, which are currently not included in MASSIVEDS.

**Discussion on inference cost.** The compute-optimal scaling study described above focuses only on the cost of training. For inference, prepending retrieved documents in context increases inference cost due to the extended context length and additional computation required for the search. On the flip side, inference cost can be reduced by switching from a larger to a smaller LM, especially since a small LM augmented with a datastore can match or outperform its larger counterparts on

---

[3]OLMO-1.7 1B is trained on up to 3T tokens and OLMO-1.7 7B is trained on up to 2T tokens.

Table 3: **Downstream and upstream performance comparison between MASSIVEDS for retrieval versus single-domain datastores with LLAMA-2 7B.** "SE" is short for StackExchange. The best performance is highlighted in **bold** and the second best is underlined. We show the diverse domain coverage in MASSIVEDS consistently improve the performance across tasks.

| Tasks | LM-Only | PubMed | MATH | peS2o | DPR Wiki | RedPajama Wiki | Books | ArXiv | SE | Github | MASSIVEDS |
|---|---|---|---|---|---|---|---|---|---|---|---|
| TQA ↑ | 64.1 | 64.5 | 65.5 | 65.6 | 72.6 | 72.9 | 70.5 | 62.3 | 64.7 | 64.2 | **77.0** |
| NQ ↑ | 26.6 | 26.7 | 26.4 | 26.9 | **34.6** | 33.8 | 28.0 | 26.4 | 27.0 | 26.4 | **34.6** |
| MedQA ↑ | 36.6 | 37.8 | 36.5 | 38.1 | 38.5 | 38.4 | **39.8** | 36.9 | 35.4 | 36.1 | 39.4 |
| MMLU ↑ | 45.8 | 46.8 | 47.5 | 47.4 | 48.3 | 48.1 | 48.3 | 45.6 | 46.2 | 45.9 | **49.3** |
| RedPajama (PPL) ↓ | 4.09 | 4.06 | 4.08 | 4.08 | 4.06 | 3.99 | 4.01 | 3.87 | 4.01 | 3.95 | **3.50** |
| S2ORC (PPL) ↓ | 7.18 | 7.05 | 7.10 | 6.71 | 7.08 | 7.11 | 7.14 | 6.64 | 7.08 | 7.11 | **6.57** |

some tasks. We also note that there is emerging work on accelerating retrieval search and designing efficient serving strategies for retrieval-based LMs, such as Cao et al. (2023). We leave a study of inference-compute-optimal scaling to future work.

# 5 Analysis

## 5.1 Effects of Data Composition

**Finding 6: MASSIVEDS matches or outperforms single-domain datastores.** The default setting in prior work is to use a single-domain datastore that is in-distribution to the downstream task. In practice, however, it is often difficult to determine and curate a datastore that is perfectly in-distribution for a downstream task, and even if we can, it limits the generality of the retrieval-based model to that task.

In Table 3, we compare MASSIVEDS with single-domain datastores. MASSIVEDS significantly outperforms these in-domain datastores on language modeling, as well as TQA and MMLU, while matching performance on NQ and MedQA.[4] In Figure 5, we show that the retriever tends to retrieve from the relevant domain even in the presence of out-of-domain data in the datastore: for NQ, it retrieves relatively more frequently from Wikipedia and web sources, whereas for MedQA, it retrieves more frequently from scientific papers from peS2o (Soldaini & Lo, 2023). Thus, the retriever can maintain robustness to out-of-domain data in the datastore; this aligns with similar findings on kNN-LM (Khandelwal et al., 2020), another type of retrieval-based LM, in Shao et al. (2023). Overall, these results show that retrieving from broad datastores like MASSIVEDS can simultaneously improve performance across multiple domains, paving the path towards general-purpose retrieval-based models.

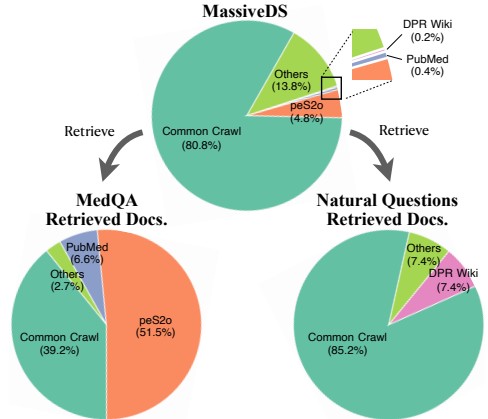

Figure 5: **Retrievers tend to retrieve from relevant domains.** We plot the domain composition of MASSIVEDS vs. the top-1 retrieved documents for evaluation examples from MedQA and NQ. The retriever retrieves more frequently from domains that are relevant to the evaluation examples.

## 5.2 Effects of Reranking

Retrieving the most relevant documents from a large-scale datastore remains a challenging problem. To study how improving the retrieval process impacts datastore scaling trends, we first retrieve 500 documents from CONTRIEVER, rerank them using a more computationally expensive model (Ram

---

[4]NQ answers are annotated based on data from a 2018 snapshot of Wikipedia, so the Wikipedia datastore is exactly the right datastore for this task.

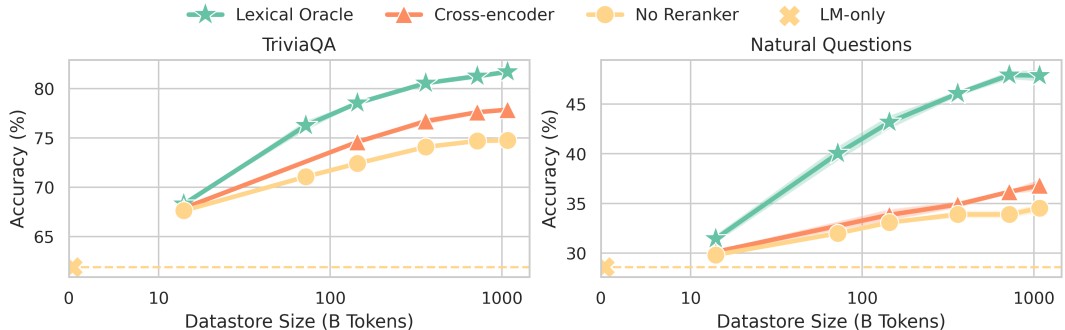

Figure 6: **Scaling trends on TriviaQA and NaturalQuestions using different rerankers (Section 5.2).** "Lexical Oracle" represents the oracle reranker that reorders documents based on lexical overlap with the ground-truth answer. "Cross-encoder" represents a neural reranker which uses a cross-encoder model. Both the oracle lexical reranker and the neural reranker boost scaling trends, indicating the potential improvement space by enhancing the retrieval quality.

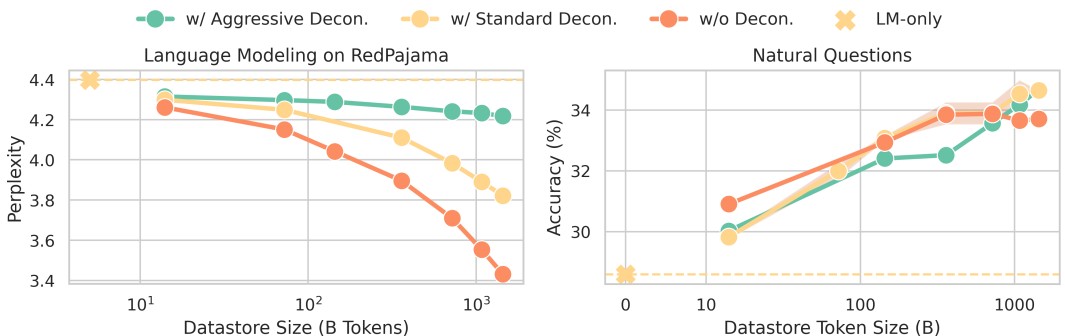

Figure 7: Ablation study on data decontamination. 'Aggressive Decon.' removes a document as long as it has an 8-gram (i.e., 1.5% of the answer length) continuous overlap with the answer. 'Standard Decon.'—our default setup—removes a document when it either has a 32-gram (i.e., 6.2% of the answer length) continuous overlap or an 80%+ Jaccard similarity with the answer. We find decontamination impacts the language modeling performance a lot but not the downstream task.

et al., 2023; Sachan et al., 2022), and take the final top-3 reranked documents for evaluation. Specifically, we use a **cross-encoder reranker**, which encodes a concatenation of a query and document and returns a similarity score (Nogueira & Cho, 2019). We choose MINI-LM-L12 V2, a BERT-based cross-encoder[5] that is trained for passage ranking, following Izacard et al. (2022). Additionally, we use a **lexical oracle reranker**, which uses the gold answer, as an upper bound on the potential benefit realizable by a better reranker or retriever for knowledge-intensive question-answering. The oracle reranker scores each document based on whether the gold answer is included in the document and if not, the fraction of unigram overlap between the document and the answer.

Figure 6 reports scaling trends on TQA and NQ of RIC-LM with Llama2-7B. While the cross-encoder-based reranker improves performance on TQA and NQ, a notable gap persists between the oracle reranker and the cross-encoder-based reranker. These suggest that improving either retrieval or reranking can further boost the scaling performance of retrieval datastores. Improving the retriever for more reasoning-heavy tasks such as MMLU and MedQA remains an open problem (BehnamGhader et al., 2022) that we leave to future work.

### 5.3 Effects of Datastore Filtering

**Data decontamination.** Data decontamination is a crucial concern when evaluating LMs, especially in retrieval-based LMs that can retrieve the test data verbatim during inference (Borgeaud et al., 2022). By default (Section 4), we perform decontamination by filtering documents with 80+% 13-gram

---

[5]`https://www.sbert.net/docs/pretrained-models/ce-msmarco.html`

Jaccard similarity for downstream tasks and 32-gram longest sequence overlap for perplexity, which we call **standard decontamination**. Prior work such as RETRO (Borgeaud et al., 2022) only used 80+% 13-gram Jaccard similarity for decontamination. However, we find the additional 32-gram longest sequence overlap decontamination is critical for removing near-identical documents.

To study the impact of varying levels of data decontamination, we compare standard decontamination with two additional methods: (1) **no decontamination** and (2) **aggressive decontamination**, which uses 8-gram longest sequence overlap for all tasks. This is a strict filter, as 8-gram overlap occurs frequently even when documents are not nearly identical.

Figure 7 reports the performance of the LLAMA-2 7B model on language modeling and the Natural Questions dataset using different decontamination methods. The scaling trend shows significantly better language modeling performance without decontamination, which worsens with more aggressive decontamination methods. This suggests that the benefits in language modeling primarily arise from lexical overlap. However, retrieval continues to benefit language modeling performance even after aggressive decontamination—where no more than 8 continuous words overlap—indicating that semantically similar retrieved documents with minimal lexical overlap can still enhance language modeling. Decontamination does not significantly affect NQ performance, likely because there is less contamination of NQ in the datastore. Interestingly, decontamination decreases performance with smaller datastores, but improves final performance at larger scales.

**Data quality filtering.** In Appendix E.2, we study the impact of data quality filtering on MASSIVEDS, where we consider global data deduplication and a combination of 3 filters adapted from DOLMA (Soldaini et al., 2024): whitespace filter; language filter, and alphanumeric filter. We find deduplication is helpful to minimizing saturation as the datastore scales on NQ; intuitively, subsampling with higher $p$ increases the chance of seeing more duplicates. In addition, we observed that DOLMA quality filters have a relatively limited effect. We hypothesize this is because the data sources we used in MASSIVEDS, such as RedPajama, have already gone through similar quality filtering processes and may not benefit much from applying additional filtering.

# 6 Limitations and Discussion

We conclude by discussing limitations and future directions. First, while our pipeline allows us to study datastore scaling efficiently, our experiments are still limited by our available compute. In particular, our compute-optimal scaling studies are limited to model families like OLMo and Pythia that release intermediate model checkpoints, since full pretraining runs exceed our budget constraints. Similarly, we conduct the full scaling study with a single retriever, as changing the retriever necessitates re-indexing the entire datastore. It remains unclear how changes in the size and architecture of the retriever affect datastore scaling trends.

Second, although MASSIVEDS is large in size, it might still lack high-quality data for improving performance on more complex, reasoning-heavy tasks such as MMLU and MedQA. Future work could study the effect of extending MASSIVEDS to more varied and higher quality data sources.

Lastly, our downstream evaluations are mostly on question-answering tasks whose outputs are either predefined choices or short form generations. We defer the evaluation on more tasks such as long-form generation and mathematical reasoning to future work.

Despite these limitations, our research shows that increasing the scale of data available at inference time can improve model performance, at lower training cost, on language modeling and a variety of downstream tasks. We expect that future work on improving retrieval-based models with large-scale datastores will lead to even larger improvements: for example, our analysis suggests that further improving the retrieval process, either through better retrievers or rerankers, could have a significant impact. We also expect scaling up the datastore to be one way to achieve extremely long-context modeling (e.g., retrieving information from a trillion-token context) and test-time scaling (e.g., spending more test-time compute to retrieve helpful information from a large datastore).

## Acknowledgements

We thank Hannaneh Hajishirzi, Scott Yih, Ian Magnusson, Matt Jordan, and Rui Xin for insightful discussions. We thank Stella Li for proofreading. PWK is supported by the Singapore National Research Foundation and the National AI Group in the Singapore Ministry of Digital Development and Innovation under the AI Visiting Professorship Programme (award number AIVP-2024-001).

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

# Appendix

# A  Datastore Construction Pipeline

The MASSIVEDS pipeline entails running the following operations in order (Figure 2):

1. **Distributed indexing.** We split the documents from each domain into separate shards and construct an index for each shard. Distributing the index allows us to parallelize indexing and retrieval processes, and to study different combinations of data sources more easily.

2. **Distributed document retrieval.** For each test query, we search for the top-$K$ documents over each shard index in parallel. The searched results are first merged within each domain. We cache the per-domain searched results for data composition analysis.

3. **Domain merging.** For a target combination of domains in the datastore, we merge the top-$K$ documents from the target domains for that query. This leads to a merged pool of $DK$ retrieved documents, where $D$ is the number of target domains. From this pool, we re-select the top-$K$ documents.

4. **Data filtering and reranking.** We then apply the data filtering and reranking steps, as described above, to only the top-$K$ documents (for each test query). This allows us to experiment with different data filters and rerankers without having to rerun indexing or retrieval; moreover, we only need to filter/rerank the retrieved results instead of the whole datastore, and we do not need to repeat these processes for different subsampling ratios.

5. **Data subsampling.** We subsample the filtered and reranked documents for each test query. Specifically, for a particular subsampling ratio $p$, we select each document for inclusion i.i.d. with probability $p$. To try different subsamples (based on different random seeds), we only need to restart the pipeline from this step, and we can reuse computation from all previous steps.

6. **Evaluation.** We use the top-$k$ documents for each test query and prepend these documents in front of the query and few-shot examples for evaluation.

In practice, we position the reranking step after data subsampling to reduce the number of documents that require reranking. The commutativity of reranking and subsampling is demonstrated in Appendix A.5. Furthermore, we collect the top-$K'$ documents (where $K' < K$) from the deduplicated and decontaminated top-$K$ set for reranking. We set $K' = k$ when reranking is not applied. Next, we describe each step of our efficiency-oriented datastore construction pipeline in detail and demonstrate the equivalence between our MASSIVEDS pipeline and the naive pipeline. Below is a table of notations for easy reference.

| Symbol | Description | Symbol | Description |
|---|---|---|---|
| $\theta$ | Parameters of the reader model | $\phi$ | Parameters of the retriever model |
| $k$ | Number of documents for evaluation | $K$ | Number of documents before deduplication |
| $K'$ | Number of documents for reranking | $\mathcal{D}$ | Retrieval datastore |

## A.1  Distributed Indexing

Our pipeline starts with raw data that is split into fixed-size documents. For each document, we run one forward pass over it with a retriever model $\phi$ and save the final-layer representation as an embedding for that document. We store the embeddings of all documents in the datastore for similarity-based search. In practice, we split the documents into separate shards and embed each shard in parallel. As we use the uncompressed embedding for retrieval searches, our indexing does not require additional operations over the saved embeddings. We treat each shard of embeddings as one sharded index. This indexing step is executed only once over all data, while the subsequent steps are performed for each query at inference. For simplicity, we describe the subsequent steps for a single query in the sections below.

## A.2  Distributed Document Retrieval

At inference, we first retrieve the top-$K$ documents from each sharded index in parallel to accelerate the search process. Specifically, we convert the query into an embedding and compute the inner-product similarity scores between this query embedding and every document embedding. We then rank the documents from each shard based on these similarity scores and collect all top-$K$ documents for subsequent merging.

## A.3 Domain Merging

Assuming we have $m$ sharded indices, we merge the $m$ sets of top-$K$ retrieved documents from all indices based on their similarity scores to obtain the final top-$K$ documents. This process is formally defined below as $m$-*sharded distributed element-wise top-$K$ retrieval*. We demonstrate that it is equivalent to directly retrieving the top-$K$ documents from a single index built on all data.

Formally, we introduce two definitions: *element-wise top-$K$ retrieval* and $m$-*shard element-wise top-$K$ retrieval*, the latter of which is what our pipeline uses.

**Definition A.1** (Element-wise top-$K$ retrieval). *Assume a datastore of $N$ documents: $\mathcal{D} = \{d_1, \cdots, d_N\}$. Given a query $q$, element-wise top-$K$ retrieval uses a retriever $g_\phi$ to compute the similarity score $s_i = g_\phi(q, d_i)$ of the query and each document $d_i$ independently, and then returns the documents with the top-K highest retrieval scores.*

**Definition A.2** ($m$-shard distributed element-wise top-$K$ retrieval). *Consider a sharding strategy that splits the datastore into $m$ shards $\{\mathcal{D}_1, \cdots, \mathcal{D}_m\}$, such that each shard contains a disjoint subset of documents $\mathcal{D}$ (i.e., $\mathcal{D}_1 \cup \mathcal{D}_2 \cup \cdots \cup \mathcal{D}_m = \mathcal{D}$; $\mathcal{D}_i \cap \mathcal{D}_j = \emptyset$, when $i \neq j$). In $m$-shard distributed element-wise retrieval, we fetch the top-$K$ documents from each shard in parallel (if there are fewer than $K$ documents in a shard, then all documents in the shard are returned). Then, we merge these $mK$ documents and return the top-$K$ highest-scoring documents from the merged pool.*

$m$-shard distributed element-wise top-$K$ retrieval is equivalent to retrieving from a single unsharded index ($m = 1$).

**Lemma A.1.** *Changing $m$, the number of shards used in distributed element-wise top-$K$ retrieval, does not impact the final retrieved results.*

*Proof.* Let the top-$K$ documents obtained by the element-wise top-$K$ retrieval and $m$-shard distributed element-wise top-$K$ retrieval be $\mathcal{D}_K$ and $\mathcal{D}'_K$, respectively. Since a document that is ranked in the top-$K$ across all documents must be ranked in the top-$K$ within any individual shard, we have $\mathcal{D}_K \subseteq \mathcal{D}'_K$. Because $\{\mathcal{D}^1, \cdots, \mathcal{D}^m\}$ is a disjoint union of $\mathcal{D}$, for any document $d$ in $\mathcal{D}'_K$, there are no more than $K - 1$ documents in the $m$ shards, i.e., all documents, that have higher scores than $d$. Then we have $d \in \mathcal{D}_K$. Therefore, $\mathcal{D}'_K \subseteq \mathcal{D}_K$. Finally, we have $\mathcal{D}_K = \mathcal{D}'_K$ for any choice of $m \in \mathbb{N}^+$. $\square$

In our experiments, we use an indexing method [6] that computes the similarity score between the query and every document, independent of other documents, i.e., element-wise retrieval. Therefore, our method for distributed search is equivalent to building and retrieving from a single non-distributed index.

## A.4 Data Filtering and Reranking

Our pipeline includes two basic data processing steps[7]: data deduplication and decontamination. These steps are applied post-hoc on the retrieved documents. Reranking, which is optional, is used to enhance the quality of retrieval. Detailed descriptions of deduplication, decontamination, and optional reranking are provided in this section.

### A.4.1 Post Hoc Datastore Deduplication

Although REDPAJAMA (Computer, 2023), the pretraining corpora used to build MASSIVEDS, has already been deduplicated, we still noticed many duplicates in the retrieved results, particularly from the web domain. This is because Computer (2023) only performs local deduplication within each data shard; globally, many duplicates between shards remain.

---

[6] We use INDEXFLATIP implemented in FAISS: `https://github.com/facebookresearch/faiss/wiki/Faiss-indexes`.

[7] Additional data quality filters, such as those from Dolma Soldaini et al. (2024), are discussed in our analysis section §5.3. These filters are applied at the same stage as deduplication.

There are two widely adopted approaches for deduplication over a large-scale dataset: the first uses a *Bloom filter*[8] and the second uses *MinHash*[9]. In short, the Bloom filter approach is typically used to remove exact duplicates and can also remove near duplicates; the MinHash approach is used to remove near duplicates detected based on n-gram Jaccard similarity. Running global deduplication over 1.4 trillion tokens using a Bloom filter requires 1.5TB RAM memory and 933 CPU hours[10]. The MinHash approach requires even more memory and CPU hours than the Bloom filter approach. Therefore, deduplicating over the entire raw text of the datastore is computationally expensive, particularly when it needs to be repeated with every experimental modification.

To get around this, we add a *post-hoc* MinHash deduplication step which is applied to a large pool of top-$K$ retrieved documents. Following Computer (2023), a document pair is a duplicate if its 13-gram Jaccard similarity score is at least 80%. Note that deduplication on the retrieved results obtained from the entire corpus is an affordable alternative to running global deduplication. The only risk is that we may not have enough documents for reranking (requiring $K'$ documents) or evaluation (requiring $k$ documents) after deduplication. Therefore, we choose a large $K$ ($K \gg K'$ and $K \gg k$), i.e., $K = 1000$, to mitigate this risk.

The original implementation of MinHash deduplication skips the chunks with less than 13 grams. Qualitatively, after deduplicating with 13-gram Jaccard similaritily, we still find many short duplicates or nonsensical phrases under 13 words in the deduplicated data pool. Thus, **we remove all documents with less than 13 words as well at the deduplication step**.

### A.4.2  Post Hoc Datastore Decontamination

Expanding the datastore to the Internet-scale incurs the risk of data contamination. Similar to global deduplication, running pre-decontamination on the entire datastore against every possible evaluation set is inconvenient and computationally expensive. Therefore, we implement a post-hoc decontamination method to remove contamination from the retrieved documents instead of the large datastore. For downstream tasks, we compare the 13-gram Jaccard similarity between each question and the retrieved document, and remove the documents with a high similarity score that is at least 80%. For language modeling (Figure 8), we adopt a stricter approach: we calculate both the 13-gram Jaccard similarity and the longest sub-sentence between the document and the answer, marking the document as contaminated if it either has at least 80% 13-gram similarity or contains a continuous 13-gram overlap with the answer. We split sentences into grams based on whitespace.

We show in Lemma A.2 that our method for post-deduplication and post-decontamination is equivalent to running deduplication and decontamination on the raw data prior to retrieval.

**Lemma A.2.** *Running post hoc exact deduplication and decontamination over the top-$K$ retrieved documents before taking the top-$K'$ (where $K' \leq K$) documents is equivalent to retrieving the top-$K'$ documents from a deduplicated and decontaminated datastore.*

*Proof.* Given a datastore $\mathcal{D}$, let the deduplicated datastore as $\mathcal{D}'$. Denote the top-$K$ documents from $\mathcal{D}$ as $\mathcal{D}_K$ and the top-$K'$ documents $\mathcal{D}'$ as $\mathcal{D}'_{K'}$. Then, denote the top-$K'$ documents retrieved from $\mathcal{D}_K$ as $\mathcal{D}_{K'}$. Since both $\mathcal{D}'K'$ and $\mathcal{D}K'$ contain the top-$K'$ ranked documents from all those retained after deduplication or decontamination, and since the removal of data is deterministic, we have $\mathcal{D}'K' = \mathcal{D}K'$. □

For approximate deduplication, there might be edge cases where the two documents are near duplicates but only one of those documents is retrieved in the top-K results. However, the property that none of the final top-k retrieved results are (near-)duplicates of each other will still hold.

### A.4.3  Reranking

Reranking is optionally applied to enhance retrieval quality in our pipeline. Given the top-$K$ retrieved documents, we apply a stronger retrieval model to embed each document and the query,

---

[8]A typical implementation is `https://github.com/allenai/bff/tree/main`.

[9]A typical implementation is `https://ekzhu.com/datasketch/`.

[10]The requirement of Bloom Filter deduplication is estimated using the implementation by AI2: `https://github.com/allenai/bff/tree/main` on our hardware setup.

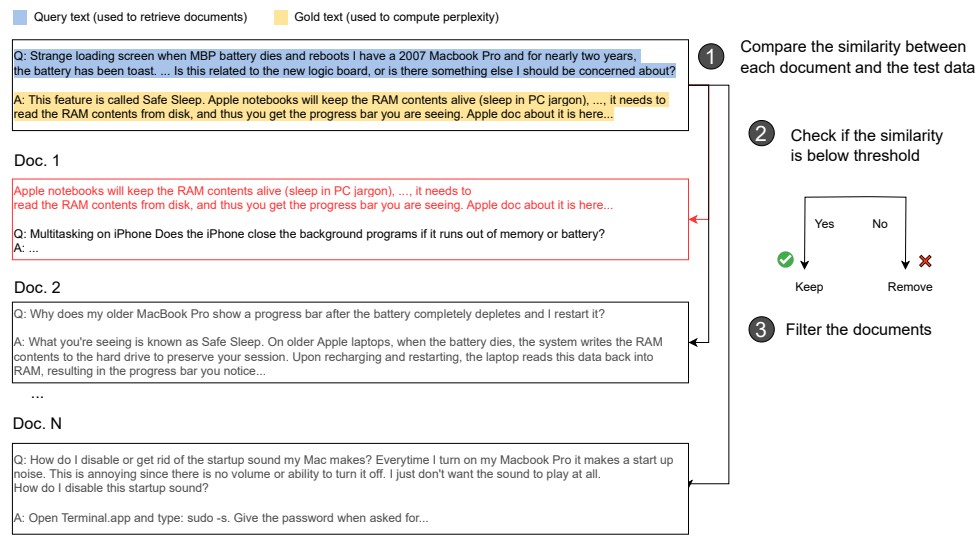

Figure 8: The post-hoc decontamination process for perplexity evaluation.

and recalculate the similarity scores to reorder the documents. Since reranking is an element-wise operation, which we demonstrate to be commutable with subsampling in the next section, we apply reranking after subsampling in practice.

### A.5 Data Subsampling

To study how model behavior changes with datastore scale, we must subsample datastores of different sizes (i.e., different sampling ratios) and with different random seeds. We first describe a naive way of subsampling from a large datastore (Algorithm 1) and show that it is computationally expensive to scale the datastore in this manner. We then introduce our efficient datastore subsampling strategy (Algorithm 2) to make datastore scaling more affordable. We also provide theoretical justification to show that our method is equivalent with high probability to naive datastore subsampling, and additionally provide a computational comparison of the two approaches. Finally, we discuss the commutativity of subsampling and other operations, demonstrating that users can flexibly reorder some of these operations in practice.

First, we define a few functions used in our algorithm.

**Function SUBSAMPLE($\mathcal{D}, p, s$).** Given a data corpus $\mathcal{D}$ with $N$ elements, we sample each element *i.i.d.* following a $\mathrm{Bernoulli}(p)$ distribution with probability $p$, and a random seed $s$ set such that for any given $(p, s)$, the same $x$ will always either be sampled or not. The total number of sampled data follows $\mathrm{Binomial}(N, p)$.

**Function SEARCH($q, \mathcal{D}, k$).** Given a query $q$ and a datastore $\mathcal{D}$, this function returns the $k$ documents from the datastore with the highest similarity to $q$.

**Function GETTOP($\mathcal{D}, k$).** This function takes in a descending ordered list $\mathcal{D}$ and returns the first $k$ elements.

A naive way of studying datastore scaling is to subsample from the raw data for every combination of subsampling ratio and random seed, as Algorithm 1 shows. However, this approach is computationally inefficient as it would entail repeatedly running subsampling over the raw data, building an index, and running retrieval search.

To affordably study datastore scaling trends, we propose an efficient way to subsample over the datastore. Instead of subsampling from the raw data, for each query, we first retrieve the top-$K$ documents from the entire datastore, and then subsample from this set. We then take the top-$k$ from

---

**Algorithm 1** Naive implementation of datastore scaling

---

1: **Input:** Data corpus $\mathcal{D}_N$, a list of subsampling ratios $P$, a list of random seeds $S$, a list of queries $Q$

2: **Output:** Retrieved top-$k$ documents $[\mathcal{D}_k^{(q,p,s)}|\, q \in Q, p \in P, s \in S]$

3:

4: **function** MAIN($\mathcal{D}_N, P, S, Q$)

5:     **for** $s \in S$ **do**

6:         **for** $p \in P$ **do**

7:             $\mathcal{D}_{pN}^s \leftarrow$ SUBSAMPLE($\mathcal{D}_N, p, s$)                       ◁ Complexity: $O(N)$

8:             Build an index over $\mathcal{D}_{pN}^s$                ◁ Complexity: $O(pNM)$[11]

9:             **for** $q \in Q$ **do**

10:                 $\mathcal{D}_k^{(q,p,s)} \leftarrow$ SEARCH($q, \mathcal{D}_{p|\mathcal{D}|}^s, k$)          ◁ Complexity: $O(pN)$

11:             **end for**

12:         **end for**

13:     **end for**

14:     **return** $[\mathcal{D}_k^{(q,p,s)}|\, q \in Q, p \in P, s \in S]$

15: **end function**

---

---

**Algorithm 2** Our efficient datastore scaling implementation

---

1: **Input:** Data corpus $\mathcal{D}_N$, a list of subsampling ratios $P$, a list of random seeds, a list of queries $Q$, number of intermediate retrieved documents $K$ ($K \ll N$)

2: **Output:** Retrieved top-$k$ documents $[\mathcal{D}_k^{(q,p,s)}|\, q \in Q, p \in P, s \in S]$

3:

4: **function** MAIN($\mathcal{D}_N, P, S, Q, K$)

5:     Build an index over $\mathcal{D}_N$                             ◁ Complexity: $O(NM)$[12]

6:     $\{\mathcal{D}_K^q|\, q \in Q\} \leftarrow$ SEARCH($Q, \mathcal{D}_{p_iN}^s, K$)         ◁ Complexity: $O(N|Q|)$

7:     **for** $s \in S$ **do**

8:         **for** $p \in P$ **do**

9:             **for** $q \in Q$ **do**

10:                 $\mathcal{D}_{pK}^{(q,p,s)} \leftarrow$ SUBSAMPLE($\mathcal{D}_K^q, p, s$)        ◁ Complexity: $O(K)$

11:                 $\mathcal{D}_k^{(q,p,s)} \leftarrow$ GETTOP($\mathcal{D}_{p_iK}^{(q,p,s)}, k$)      ◁ Complexity: $O(1)$

12:             **end for**

13:         **end for**

14:     **end for**

15:     **return** $[\mathcal{D}_k^{(q,p,s)}|\, q \in Q, p \in P, s \in S]$

16: **end function**

---

the pool of $K$ documents for final evaluation. Algorithm 2 shows the pseudocode for our proposed approach.

**Computation comparison.** The complexity of naive datastore subsampling is $O((1 + M + |Q|)|P||S|N)$, where $M$ is the number of parameters of the retriever model, $Q$ is the number of queries, $|P|$ is the number of subsampling ratios, $|S|$ is the number of random seeds, and $N$ is the number of documents in the datastore. The complexity of naive datastore subsampling is dominated by $O(M|P||S|N)$ in practice, which is the cost of repetitively building the datastore for all combinations of subsampling ratios and random seeds. The complexity of our subsampling strategy is only $O(N(M + |Q|) + K|P||Q||S|)$, which is dominated by $O(MN)$, representing the cost of one-time indexing.

**Lemma A.3.** *Subsampling from the retrieved top-$K$ documents with probability $p$ and then taking the top-$k$ ($k \ll K$) from the subsampled documents (Algorithm 2) is equivalent with high probability to directly retrieving top-$k$ documents from a datastore that is subsampled from the entire raw text datastore with probability $p$ (Algorithm 1). The equivalence holds as long as there are enough $k$*

*documents left after subsampling. The chance of failure, i.e., not having enough documents left, is exponentially small in $K$.*

*Proof.* Assume a fixed random seed set in a way that whether each document in the raw data pool will be included or not is determined. If a document is subsampled in the naive approach, it will also be sampled in our approach, because this document has a retrieval score falls in top-$k$ and it is determined to be sampled. Therefore, as long as there are at least $k$ documents left after our subsampling over the top-$K$ documents, the results are guaranteed to be the same. Since the number of documents remaining after subsampling follows a binomial distribution with parameters $K$ and $p$, we can use standard tail bounds to show that the failure of not having enough documents is exponentially low in $K$. □

Lemma A.3 holds with an exponentially small probability for failure. The probability for failure is very small in practice, i.e., less than 1% if we have 1000 documents for subsampling. To guarantee equivalence, we can use a fallback mechanism where if we do not have enough documents left are subsampling, we try again with a larger value of $K$. However, in practice, we do not use this fallback as the failure rate is very low in our experiments.

**Commutability of Operations.** We next discuss the commutability between subsampling and other operations. To begin, we distinguish the data operations into two sets: *element-level operations* and *set-level operations*.

**Definition A.3** (Element-level operation). *An element-level operation is conditioned on a single element, i.e., a document in our context. For example, assigning an element-wise score to each document during reranking is an element-level operation.*

**Definition A.4** (Set-level operation). *A set-level operation refers is conditioned on a set of elements containing at least two elements. For example, deduplication is a set-level operation.*

**Lemma A.4.** *Independent element-level operations are commutable with each other. Set-level operations are not commutable.*

*Proof.* Since the results of independent element-level operations are not impacted by their order of execution, they are commutative with each other. However, this does not hold true for set-level operations, which are order-sensitive. □

We note that both merging and subsampling can be considered independent element-level operations if we regard the removed documents as being classified by either process to be masked out. As a results, operations such as data quality filters, data decontamination, reranking can be moved around before or after post-hoc merging, which made it possible for us to efficiently evaluate the effect of their variants by moving them to after retrieval and merging.

**Proposition A.1.** *Our MASSIVEDS pipeline is equivalent to the naive pipeline, as shown in Figure 2, with high probability.*

*Proof.* Lemma 1 shows that the distributed indexing and retrieval is equivalent to unsharded indexing and retrieval. Lemmas 2-4 show that the operations that we reordered between the naive pipeline and the MASSIVEDS pipeline commute without changing the returned results, with a failure probability exponential in K, where the randomness is due to subsampling. Thus, the pipelines are equivalent with high probability. □

## A.6   Evaluation

After the aforementioned operations, we select the top-$k$ documents from those retained for evaluation. We refer to the next section for a detailed evaluation setup.

# B Implementation Details

## B.1 Pipeline

**Sharding.** We split the raw data of each domain into $m$ shards, with $m$ determined based on domain size. Specifically, $m = 32$ for each time slice of CommonCrawl and C4, and $m = 8$ for the other domains.

**Chunking.** Prior to datastore construction, we chunk the raw data in each shard into fixed-length passages of at most 256 words each.

**Deduplication.** Deduplication is a process that removes documents with high overlap within a data pool. Following Borgeaud et al. (2022); Magnusson et al. (2023), we deduplicate the retrieved top-$K$ documents based on 13-gram Jaccard similarity between document pairs. Document pairs that share at least 80% similarity are marked as duplicates, and the document with the lower retrieval score is removed.

**Decontamination.** Decontamination is a process that removes documents that share high similarity with evaluation data. For upstream language modeling, we apply a combination of two decontamination methods: *13-gram Jaccard decontamination* and *32-gram longest decontamination*. Specifically, 13-gram Jaccard decontamination computes the 13-gram Jaccard similarity between the document and the answer, and the document is removed if it shares at least 80% similarity score with the answer. 32-gram longest decontamination removes documents that overlap with the answer by a contiguous sequence of at least 32 tokens. Note that we use an 512-token answer to compute perplexity, so the decontamination ratio is 0.0625 of the answer length. For downstream tasks, we apply 13-gram Jaccard decontamination and remove retrieved documents with at least 80% similarity to the test data.

**Hyper-parameters.** Our pipeline has three main hyper-parameters: $k$, $K$, and $p$. $k$ is the number of documents used for evaluation. $K$ is the number of documents retrieved before subsampling. $p$ is the subsampling ratio which controls the size of the datastore. We consider $k = 3$, $K = 1000$, and $p = [0.01, 0.05, 0.1, 0.25, 0.5, 0.75, 1]$. We also set different random seeds for the subsampling process. We run each subsampling with three seeds $(100, 101, 102)$ to obtain the confidence intervals in our scaling analyses. We provide a lookup table of the tail bound for our subsampling algorithm's failure to provide enough documents for evaluation in Table 4. This indicates that our setup is approximately equivalent to performing expensive subsampling on the raw data first based on Proposition A.3.

Table 4: Lookup table of the tail bound for a binomial distribution $Binomial(K, p)$ with at least $m = 3$ number of successes.

|            | $p = 0.01$ | $p = 0.05$ | $p = 0.1$ | $p = 0.25$ | $p = 0.5$ | $p = 0.75$ |
|------------|-----------|-----------|----------|-----------|----------|-----------|
| $K = 1000$ | 0.9973    | 1.000     | 1.000    | 1.000     | 1.000    | 1.000     |

## B.2 Language Modeling Evaluation Setup

Following Baevski & Auli (2019); Khandelwal et al. (2020); Min et al. (2023a), we split evaluation data into into fixed-length chunks of 1,024 tokens, with a stride of 512 tokens. For each chunk, the first half is used as a prefix and retrieval query, and the second half as the target sequence for calculating perplexity.

## B.3 Downstream Evaluation Setup

Table 5 shows the domain, metric, and sample count for each downstream task. We evaluate all downstream tasks in a 5-shot setting and prepend the top-3 documents for retrieval-based LM evaluation. We adapt the `lm-evaluation-harness`,[13] a widely used LM evaluation suite, for downstream evaluation.

---

[13]https://github.com/EleutherAI/lm-evaluation-harness

Table 5: Downstream evaluation tasks.

| Task | Type | Domain | Metric | Sample Count |
|------|------|--------|--------|--------------|
| TQA (Joshi et al., 2017) | Open-domain QA | Wikipedia | Exact Match | 17,944 |
| NQ (Kwiatkowski et al., 2019) | Open-domain QA | Wikipedia | Exact Match | 3,610 |
| MEDQA (Jin et al., 2020) | Science QA | Medical | Accuracy | 1,273 |
| MMLU (Hendrycks et al., 2020) | Knowledge Reasoning | Varied | Accuracy | 14,042 |

**Prompt format.** We tested two variants of prompt format. The first format starts with the few-shot examples and is followed by retrieved documents. The second format starts with the retrieved documents followed by the few-shot examples. We found the LM can learn the few-shot pattern better when the few-shot exmaples are closer to the question, leading to a superior performance than the other way. Therefore, we stick to the second format through the paper.

### B.4 Compute-Optimal Scaling Setup

**Intermediate checkpoints.** Thanks to prior works that release intermediate checkpoints, we can study the computational scaling behaviors approximately without pretraining LMs from scratch. In particular, we consider the intermediate checkpoints provided by Pythia (Biderman et al., 2023) and OLMo (Groeneveld et al., 2024). Pythia provides checkpoints of 9 sizes trained on up to 300B tokens from the Pile (Gao et al., 2020). We consider Pythia models that have at least 1B parameters, i.e., PYTHIA-1B, PYTHIA-2.8B, PYTHIA-6.9B, and PYTHIA-12B, in favor of their capability of handling complex downstream tasks. Additionally, we consider OLMO-1.7-1B and OLMO-1.7-7B which are trained on 3T and 2T tokens from Dolma (Soldaini et al., 2024), respectively. For Pythia, we use checkpoints trained on 1/30, 1/15, 1/10, 1/5, 1/4, 1/3, 1/2, and all of the full corpus. For OLMo, which only has models of two sizes, we select additional checkpoints trained on 1/50, 1/40, 1/20, 1/9, 1/8, 1/7, and 1/6 of the full corpus.

**FLOPs calculation.** Following (Kaplan et al., 2020; Hoffmann et al., 2022; Gadre et al., 2024), we approximate the FLOPs needed for one forward pass as $\text{FLOPs}_{\text{fwd}} \approx 2ND$ and for a backward as $\text{FLOPs}_{\text{bwd}} \approx 4ND$, where $N$ is the number of parameters and $D$ is the number of tokens. Pretraining requires one forward pass and one backward pass on every token in the pretraining corpus. Denote the size of LM as $N_{\text{LM}}$ and the size of the pretraining corpus as $D_{\text{pretrain}}$. The FLOPs for pretraining can be approximated as $\text{FLOPs}_{\text{pretrain}} \approx 6N_{\text{LM}}D_{\text{pretrain}}$. Datastore construction requires one forward pass on every token in the datastore corpus during the embedding step. Denote the size of the retriever as $N_{\text{retriever}}$ and the size of the datastore corpus as $D_{\text{datastore}}$. The FLOPs for datastore construction can be approximated as $\text{FLOPs}_{\text{datastore}} \approx 2N_{\text{retriever}}D_{\text{datastore}}$. Because we use a flat index, i.e., no additional operations are required at the indexing step, the FLOPs for datastore construction equal the FLOPs for embedding. We note that other types of indexing, e.g., inverted file index (IVFADC) (Jégou et al., 2011), may require additional FLOPs during construction and fewer FLOPs at inference.

## C  Complete Datastore Scaling Results

In this section, we supplement the datastore scaling performance of PYTHIA and OLMO models, in addition to the LLAMA models shown in Figure 3. Specifically, we present the datastore scaling performance for the following model families:

- **Pythia** (Biderman et al., 2023) of 4 sizes: PYTHIA-1B, PYTHIA-2.8B, PYTHIA-6.9B, and PYTHIA-12B.

- **OLMo** (Groeneveld et al., 2024) of 2 sizes: OLMO-1B and OLMO-7B.

- **Llama** (Touvron et al., 2023), which we consider LLAMA-2 7B, LLAMA-2 13B, and LLAMA-3 8B.

The complete datastore scaling results for TriviaQA, Natural Questions, MMLU, and MedQA are shown in Figures 9, 10, 11, and 12, respectively. We find retrieval benefits LMs of varied sizes across different LM families. In particular, the results show that small LMs outperform larger LMs of the

same model architectures when augmented with MASSIVEDS. Surprisingly, Pythia-1B matches Pythia-12B when augmented with only 100B tokens on knowledge-intensive tasks such as TriviaQA and Natural Questions, and it outperfoms Pythia-12B when further increasing the size of datastore.

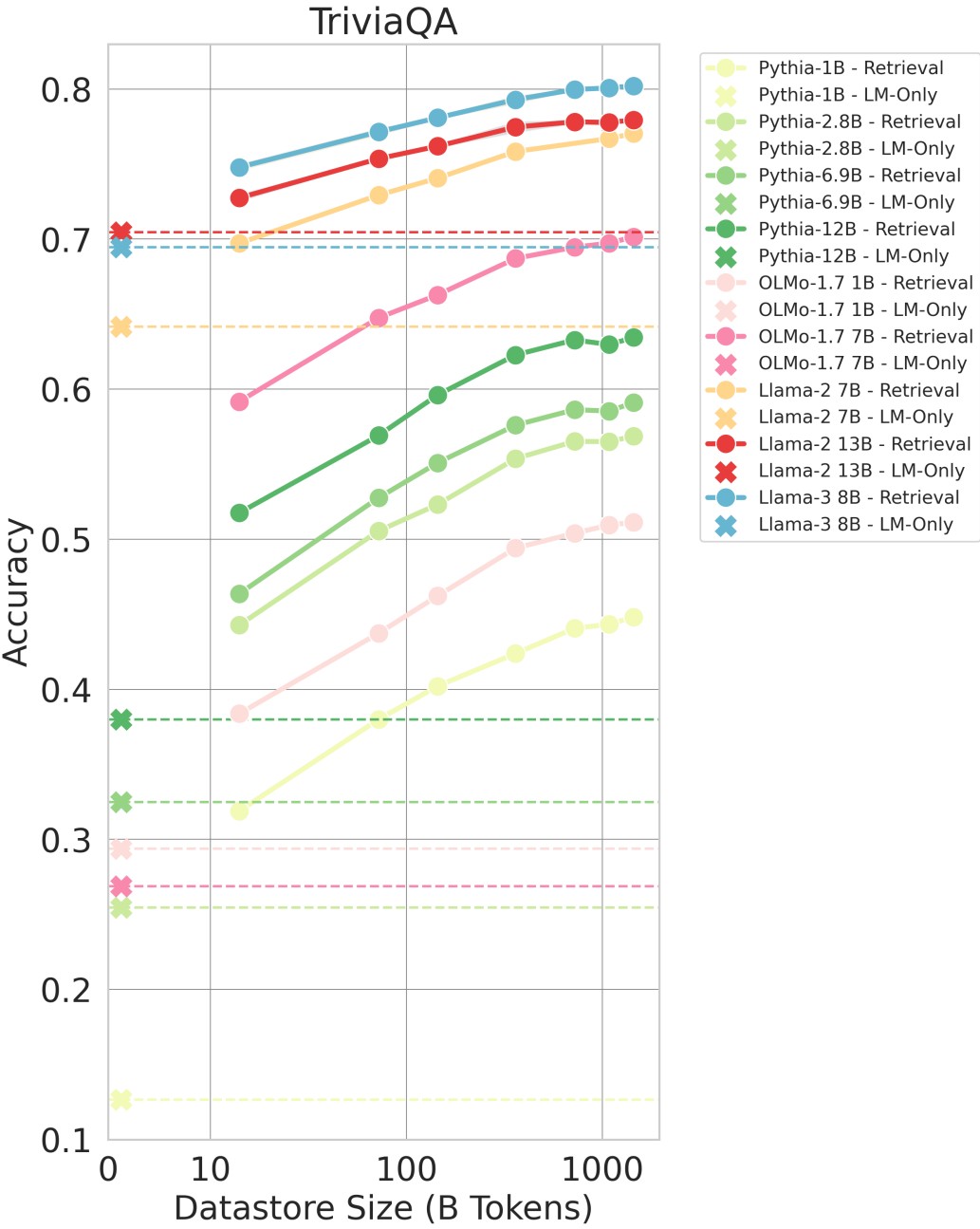

Figure 9: Complete datastore scaling performance on TriviaQA with PYTHIA OLMO and LLAMAmodels.

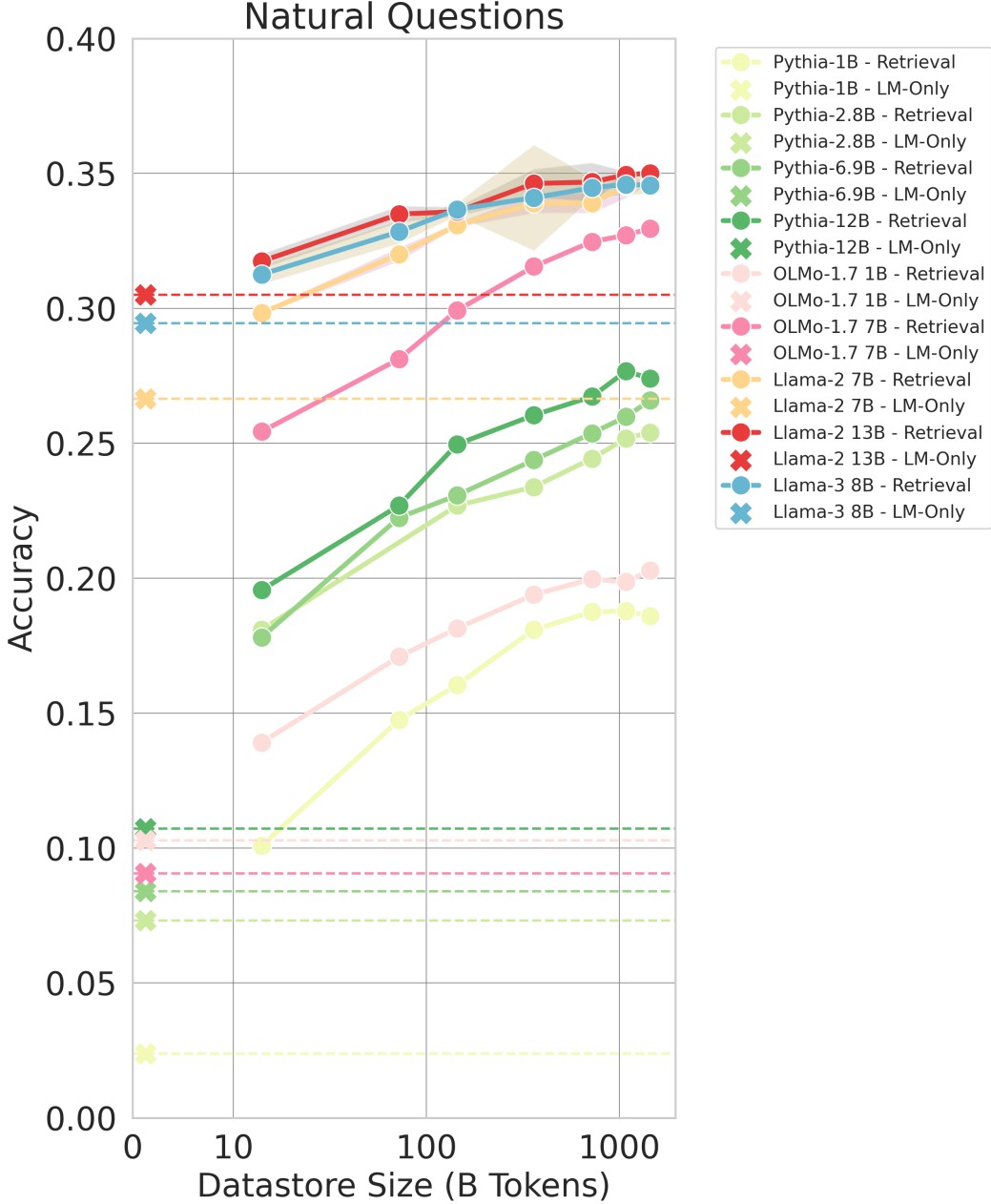

Figure 10: Complete datastore scaling performance on Natural Questions with PYTHIA OLMO and LLAMAmodels.

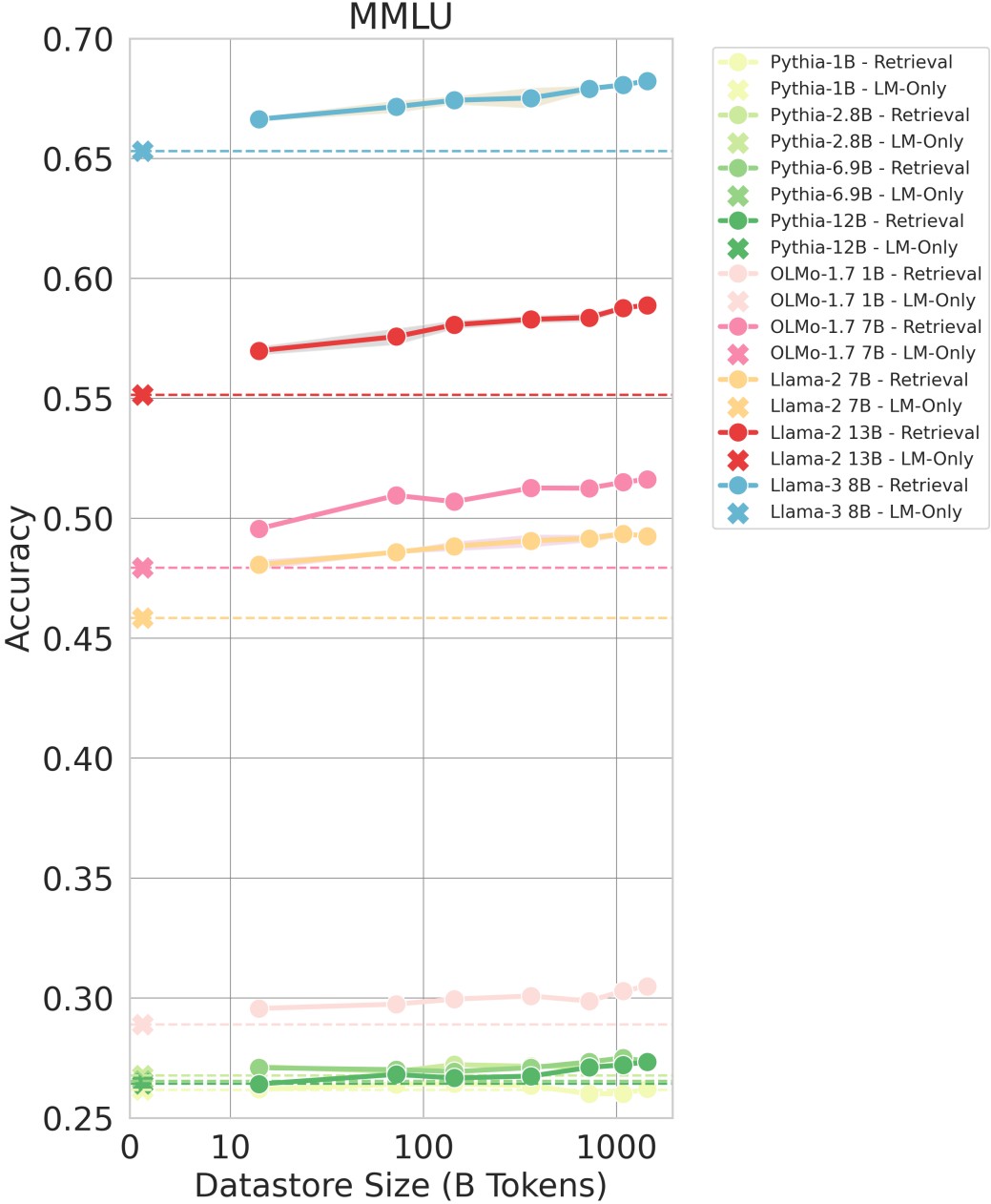

Figure 11: Complete datastore scaling performance on MMLU with PYTHIA OLMO and LLAMAmodels.

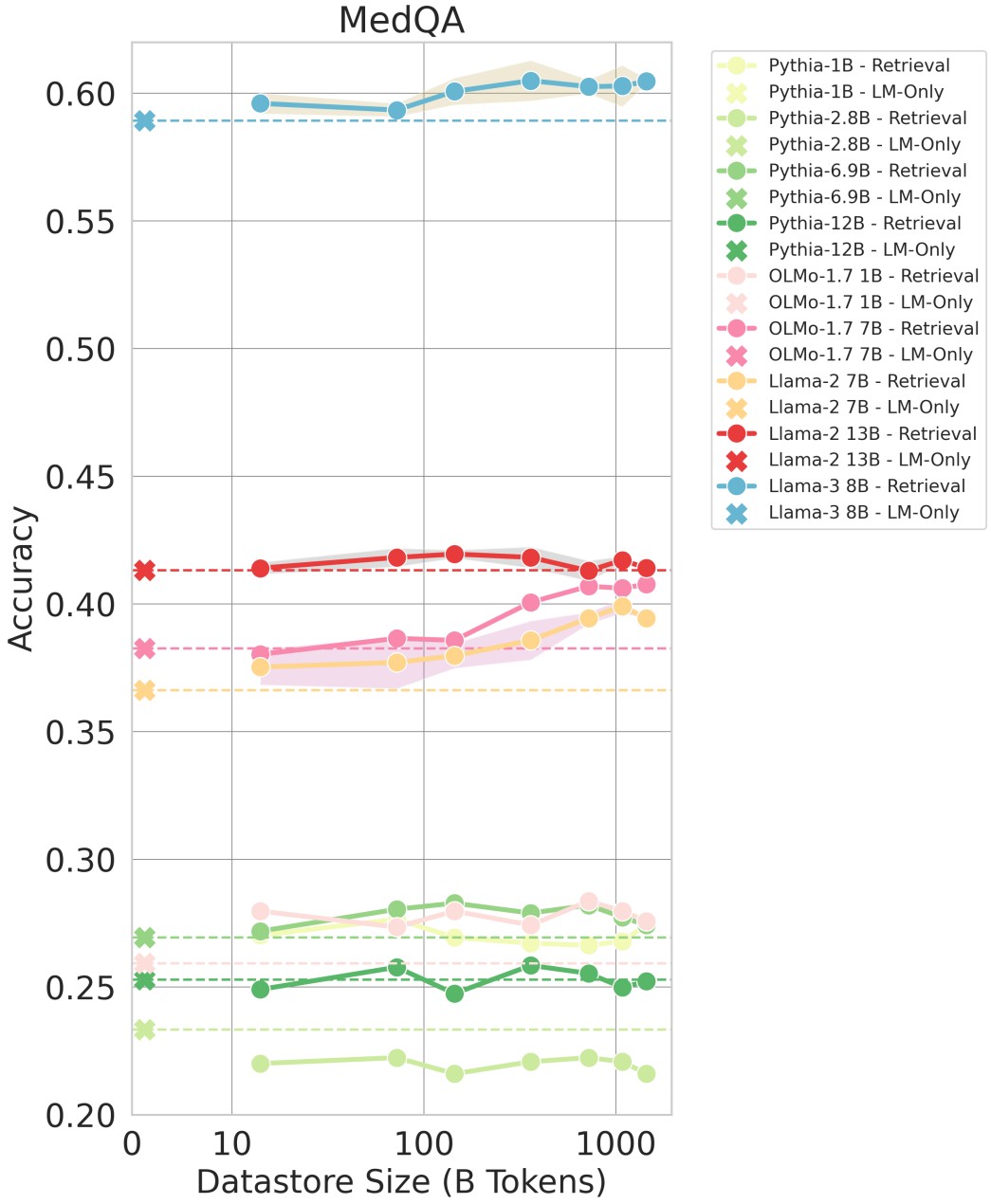

Figure 12: Complete datastore scaling performance on MedQA with PYTHIA OLMO and LLAMAmodels.

# D Discussion on the Inferior LLAMA-3 PPL Performance on RedPajama Data

We show in Figure 3 that LLAMA-3 8B has worse PPL scores than LLAMA-2 7B on RedPajama data with and without retrieval augmentation, which contradicts the intuition that LLAMA-3 8B is stronger than LLAMA-2 7B. In fact, Xiao et al. (2023) also reported worse PPL scores for LLAMA-3 8B than LLAMA-2 7B. We note that many factors can contribute to the difference in PPL evaluation, such as the pretraining data and context length. For example, LLAMA-3 incorporated a post-training process using instruction-tuning data, which aims to enhance model alignment and output quality for complex tasks but could shift performance metrics away from those optimal for simple PPL evaluations. In

addition, LLAMA-3 was trained with significantly more data— a domain (such as RedPjama) could be down-weighted in a larger corpus, leading to less memorization for this certain domain during pretraining.

# E    Additional Analysis

In this section, we provide additional analyses of the impact of retriever and data quality filtering.

## E.1    Ablation on the Retriever

In this section, we ablate the choice of retriever. Since datastore construction is expensive, we subsample 10% from the full corpus of MASSIVEDS for this ablation study. We build datastores with 3 different retrievers, CONTRIEVER-MSMARCO (Izacard et al., 2022), DRAGON-ROBERTA (Lin et al., 2023), GTR-T5-BASE (Ni et al., 2021), and evaluate on upstream perplexity and downstream tasks with them separately. A subset of the tasks, i.e., RedPjama for language modeling evaluation, and NQ and MMLU for downstream evaluation, is used to compare different retrievers. The evaluation results are shown in Table 6. We find the 3 retrievers perform similarly.

Table 6: Ablation on the retriever. We evaluate different retrievers using 10% randomly sampled MASSIVEDS. We evaluate with LLAMA-2 7B on language modeling with RedPajama data and downstream tasks Natural Questions and MMLU. The best performance is highlighted in **bold**, and the second best is underlined.

| Name | Retriever Type | Size | Perplexity ↓ | Natural Questions ↑ | MMLU ↑ |
|---|---|---|---|---|---|
| Contriever | dense | 177M | 4.2210 | 0.3321 | 0.4922 |
| DRAGON | dense | 110M | 4.2373 | **0.3399** | 0.4875 |
| GTR-Base | dense | 110M | **4.2146** | 0.3080 | **0.4934** |

We empirically find the implementations of Contriever[14] and DRAGON[15] run much faster than the sentence-transformer[16] implementations, e.g., GTR-Base [17]. As a result, we choose Contriever in our full-size scaling study with a consideration of both performance and efficiency.

## E.2    Effect of Data Quality Filtering

**Setup.**   We consider three data quality filters applied to the DOLMA (Soldaini et al., 2024) corpus: (1) a **whitespace filter** which counts the number of whitespace-separated tokens in each document, and filters out documents with counts under a manually defined threshold; (2) a **language filter** which uses a FastText (Bojanowski et al., 2017) model to detect the language of the document and filters out those with low model confidence; (3) an **alphanumeric** filter which only retains documents which contain at least one alphanumeric character, and do not contain a span of all punctuation characters (the length of the span is user-defined).

**Data deduplication.**   Removing data duplicates has proven effective for pretraining more compute-optimal language models (Lee et al., 2022). Duplicates are especially undesirable in the context of retrieval augmentation as they repeat the same information while increase the inference cost, raising a need for global deduplication. In our default setting (Section 4), we perform global datastore deduplication based on 13-gram Jaccard similarity, similar to Lee et al. (2022). Additionally, we report results without global deduplication for comparison.

Figure 13 (b) and (e) report the results on language modeling perplexity (on RedPajama) and on NQ, respectively. We find negligible impact of global deduplication in language modeling perplexity. On

---

[14]`https://github.com/facebookresearch/contriever`

[15]`https://huggingface.co/facebook/dragon-roberta-query-encoder`

[16]`https://sbert.net/`

[17]`https://huggingface.co/sentence-transformers/gtr-t5-base`

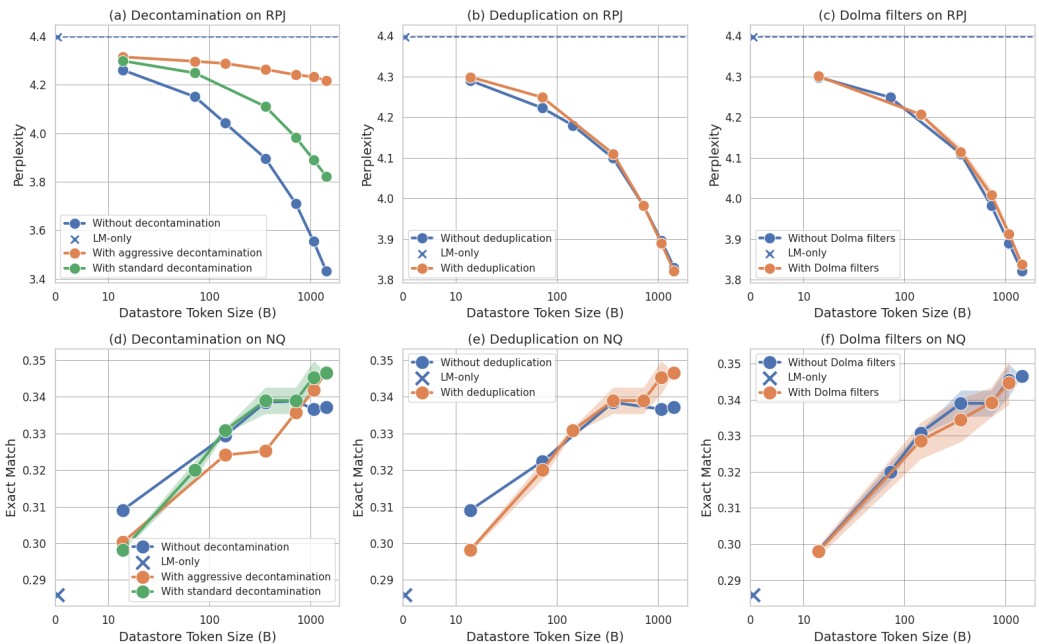

Figure 13: Analysis on the effect by deduplication, decontamination, and quality filters, from left to right (Section 5.3). The first row corresponds to language modeling on REDPAJAMA and the second row shows QA results on Natural Questions (NQ).

NQ, deduplication is crucial to minimizing saturation as the datastore scales; intuitively, subsampling with higher $p$ increases the chance of seeing more duplicates [18].

**DOLMA Quality filtering.** To study the impact of quality filtering, we consider a data filter adapted from DOLMA (Soldaini et al., 2024) which uses a combination of three filters: a whitespace filter; a language filter, and an alphanumeric filter (detailed in Appendix Section E.2).

Figure 13 (c) and (f) indicate that the quality filter has a relatively limited effect. We hypothesize that the data sources we used in MASSIVEDS, such as RedPajama, have already gone through similar quality filtering processes and may not benefit much from applying additional filtering. While not explored in this paper, recent studies indicate that more computationally expensive but higher-quality filters can further enhance pre-training (Abbas et al., 2023; Penedo et al., 2023); we suggest future work to explore such filters for potential performance improvements.

**Removing Small Chunks from MASSIVEDS** We show a few examples of the top-1 retrieved documents before and after removing the short chunks that have less than 13 words in Figure 14. Without removing these chunks, we find retriever tend to retrieve documents with verbatim text overlap to the question, but do not provide helpful information about the answer, leading to a degradation in end-task performance. Figure 15 indicates that removing short chunks can significantly improve the retrieval-based LM performance with a large datastore, which is more likely to contain short, but meaningless word chunks.

---

[18]The source corpus of MASSIVEDS has been applied moderate deduplication. For example, RedPajama preprocessed the data using several filters: `https://github.com/togethercomputer/RedPajama-Data/tree/rp_v1/data_prep`. However, we still find many duplicates in the retrieved results.

```
{
    "question": "Answer these questions:\n\nQ: when did the eagles win last super bowl?\nA:",
    "before": "Eagles won the Super Bowl.",
    "after": "As someone who lived in Philly for about five years, I agree about the city\u2019s greatness \u2014 whic
},
{
    "question": "Answer these questions:\n\nQ: who sang i ran all the way home?\nA:",
    "before": "All the Way Home",
    "after": "I Ran All the Way Home contains the vocal group's two hits: the smash title track and the minor hit Oh,
},
{
    "question": "Answer these questions:\n\nQ: who plays gram on the young and the restless?\nA:",
    "before": "on The Young and the Restless",
    "after": "Michael's dad River Lowell Baldwin played kwa Michael Gross. . Wallpaper and background images in the Th
},
{
    "question": "Answer these questions:\n\nQ: who developed the concept of total quality management?\nA:",
    "before": "Introducing total quality management.",
    "after": "Total quality management it may have been first coined in the United States by the Naval Air Systems Com
},
{
    "question": "Answer these questions:\n\nQ: who wrote cant get you out of my head lyrics?\nA:",
    "before": "\u2018Can\u2019t Get You Out Of My Head\u2019",
    "after": "the need in me I just can't get you out of my head (La,la,la La,la,la,la,la) Songwriters: Cathy Dennis /
},
{
    "question": "Answer these questions:\n\nQ: last episode of what happens to my family?\nA:",
    "before": "What happened to the family?",
    "after": "As a child, I used to love to watch the television series Columbo because at the end of almost every epi
},
{
    "question": "Answer these questions:\n\nQ: who wrote it's a long long way to pasadena?\nA:",
    "before": "Long Way",
    "after": "(see 1975 in music), and was written by Angus Young, Malcolm Young and Bon Scott. it's a long way to the
},
{
    "question": "Answer these questions:\n\nQ: who is the first wife on sister wives?\nA:",
    "before": "first wife?",
    "after": "Q: Name for relation between a man\u2019s two wives? What is the relation between the two wives of a man
},
{
    "question": "Answer these questions:\n\nQ: when does the dlc for rainbow six siege come out?\nA:",
    "before": "for Rainbow Six: Siege",
    "after": "the first-person shooter, as well as increasing some of the strategic choke-points. The DLC will enter i
},
{
    "question": "Answer these questions:\n\nQ: where does the last name galvez come from?\nA:",
    "before": "Galvez",
    "after": "The mountainous borders of Spain contain the origins of the prestigious surname Galvez. The earliest for
},
{
    "question": "Answer these questions:\n\nQ: where do you get a cashiers check from?\nA:",
    "before": "Get a Cashier\u2019s Check From Any Bank?",
    "after": "Find out where to get a cashier\u2019s check and what steps to take to get one from a bank. A cashier\u2
},
{
    "question": "Answer these questions:\n\nQ: what is the meaning of cc and bcc?\nA:",
    "before": "what CC and BCC mean and security issues.",
    "after": "Cc stands for carbon copy and it means that whoever name appears after the Cc: will get a copy of the me
},
```

Figure 14: Retrieved documents for NQ when short chunks ($<$ 13 words) are not removed. ''questions'' are inputs from NQ; ''before'' refers to the top-1 retrieved document when we do not remove short chunks; ''after'' refers to the top-1 document after a short-chunk removal step is added to our pipeline. The retriever tends to retrieve a short chunk that may have high lexical overlap with the question, but does not provide any useful information for the answer.

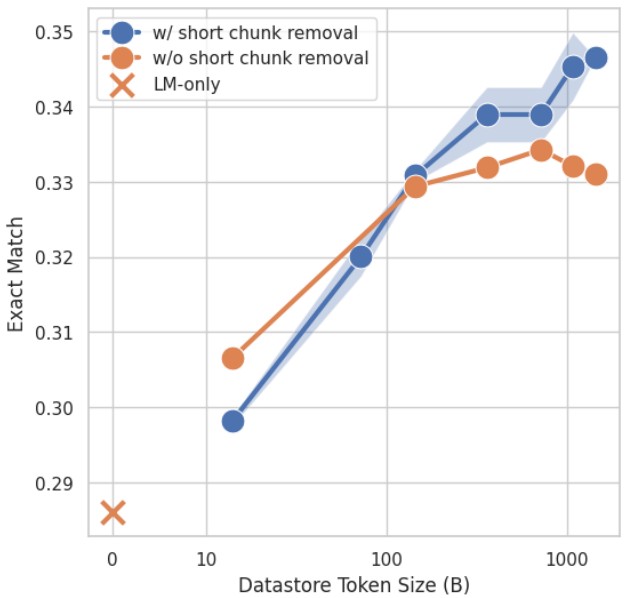

Figure 15: Comparison of NQ performance with and without the removal of short chunks from MASSIVEDS. We use LLAMA-2 7B as the reader model.

