# OpenReview forum: "Scaling Retrieval-Based Language Models with a Trillion-Token Datastore"
_NeurIPS.cc/2024/Conference — NeurIPS 2024 poster_

### Official Review · Reviewer_AKfc · 2024-07-06

**Soundness:** 3
**Presentation:** 4
**Contribution:** 3
**Rating:** 6
**Confidence:** 4

**Summary:**

The paper introduces a substantial datastore named MASSIVEDS for retrieval-in-context language models (RIC-LM). It details a comprehensive construction pipeline, with a notable deviation from the traditional datastore construction sequence. Specifically, it places indexing and retrieval at the initial stage, followed by the merging and filtering of retrieved results. The evaluation indicates that RIC-LM with MASSIVEDS, outperforms standalone language models in knowledge-intensive tasks. Additionally, the authors analyze the scaling behavior of the datastore under various configurations.

**Strengths:**

1. The exploration of how datastore configurations impact retrieval-based language models is both intriguing and significant. Some of the findings provide valuable contributions to the advancement of these systems.

2. The paper is well-written and accessible, with each step of the implementation thoroughly explained and presented.

3. The proposed datastore holds potential for related research in this field.

**Weaknesses:**

1. The paper attempts to draw critical and general conclusions, yet the scope and depth of their experiments lack the robustness required to fully support these assertions. The study focuses on the impact of various datastore configurations, such as size, filters, and decontamination strategies. Evaluations are carried out on OpenQA tasks using a retriever specifically trained for QA. However, several key questions remain unaddressed. For example, it is unclear whether the findings would remain consistent using a single data source, whether the conclusions apply to specific tasks that are distinct from broader tasks like QA, and how the interplay between different types of retrievers and language models might affect the results. Without exploring these areas, claiming a definitive trend is premature.

2. Regarding the proposed MASSIVEDS, there is a lack of a comparative analysis and comparson with other existing datastores [1, 2]. Merely assembling a large and diverse datastore does not inherently indicate a significant contribution or guarantee enhanced performance. Evaluating the proposed datastore against these established frameworks could help justify claims of superiority or uniqueness.

3. There are several major assertions may be incorrect or lack sufficient justification. For instance:
    - The claim that indexing the entire datastore initially reduces computational overhead is debatable. In practice, dense vectors can be stored and reused, and there is no need to re-index any data twice. Therefore, the upper bound of computational cost remains indexing the entire datastore. This aspect could be also alleviated significantly by considering alternative indexing strategies [3, 4].
    - The overlap between the datastore and the downstream evaluation dataset raises concerns about data leakage.  The presence of RedPajama data in MASSIVEDS (Table 2), and its use in downstream evaluation (line 191) necessitates a clearer delineation of this overlap and its impact on the results. Clarifying this could address the validity of the claim that "Retrieval is strictly beneficial for language modeling" (line 214).
    - The use of models such as TinyLlama and Llama-7B to conclude that "retrieval effectively closes the performance gap between small and large models" (line 225) appears inconclusive in the context of current advancements in LLM. Reevaluating this claim with additional models might provide a clearer picture.

Typos:
- line193 into into -> into
- line726: exmaples -> examples


[1] Soldaini, Luca, et al. "Dolma: An open corpus of three trillion tokens for language model pretraining research"

[2] Gao, Leo, et al. "The pile: An 800gb dataset of diverse text for language modeling"

[3] Li, Haitao, et al. "Constructing tree-based index for efficient and effective dense retrieval"

[4] Zhou, Jiawei, et al. "Semi-Parametric Retrieval via Binary Token Index"

**Questions:**

1. Could you elaborate on how few-shot exemplars are selected for QA tasks?

2. For downstream evalaution, how is the language model configuration determined? Is there any inherent randomness in the LM inference process, and if so, what measures are taken to mitigate its effects?

3. Following W3, have the authors verified whether the downstream evaluation dataset overlaps with the datastore? How is this ensured, and how do you analyze the extent to which observed improvements might be influenced by potential data leakage?

4. Given that many of the stages within MASSIVEDS, such as parallel indexing and aggregating retrieval results from different sources, are common practices in retrieval pipelines, how do you substantiate that the contribution of MASSIVEDS is significant and not merely an assembly of various data segments?

5. Suggestion: given the emphasis on streamlining the datastore construction pipeline, proposing multiple variants of the datastore setup (such as variations in source and chunk size) might be more advantageous than focusing solely on creating the largest and most diverse datastore. This could facilitate more comprehensive analyses and support broader conclusions for future works in this domain.

**Limitations:**

In the limitation section, the authors did acknowledge their limited analysis concerning the systematic range of retrievers and language models, which is indeed crucial. However, it could be problematic to draw broad conclusions on the datastore scaling trend from such a restricted dataset and limited scope of experiments.  A more comprehensive analysis is necessary, or alternatively, the conclusions should be more narrowly defined to ensure precision and avoid misleading interpretations.

---

> ### Author Rebuttal · Authors · 2024-08-07
>
> We thank the reviewer for acknowledging that our work makes valuable contributions to the advancement of these systems!
>
> **Weakness 1.** It remains uncertain if the findings would hold with a single data source, whether they apply to specific tasks beyond general ones like QA, and how interactions between various retrievers and language models might influence the results.
>
> **A1.** We will carefully adjust our claims in our final draft to highlight our choice of models and retrievers, and that our downstream tasks focus on QA. Given resource constraints, we chose to focus on the most widely adopted RAG designs and evaluation setups; we agree that there are many more questions to be studied, and we hope that our open-source data and code will facilitate such work in the future.
>
> On the specific comments:
>
> (1) It is hard to scale single data sources like Wikipedia to the trillion token regime, so we focus on multi-domain web-scale data; however, we compare with single in-domain data sources in Table 3.
>
> (2) We agree it would be good to study more downstream tasks beyond QA, and we will emphasize this in the discussion. We note that it is common in prior work (RePlug, Shi, et al; Atlas, Izacard, et al.) to focus on QA as well, and that we are the first to study downstream tasks at large datastore scales.
>
> (3) We have added results on new LMs (Llama3, Olmo, and Pythia families) and new retrievers (DRAGON and GTR) in Figure R1 and Table R1 in the general reply, respectively; these results are consistent with our submission.
>
> ---
>
> **Weakness 2.** There is a lack of comparative analysis and comparison of MASSIVEDS with other existing datastores [1, 2] (Dolma and Pile).
>
> **A2.** We would like to clarify that a datastore, an indexed resource ready for retrieval, differs from a pretraining corpus, which contains only raw data, like Dolma and Pile. Constructing a datastore from a pretraining corpus involves data cleaning, chunking, embedding, and indexing, which take substantial effort, and as such there are no comparable open-source datastores to MassiveDS.
>
> That said, the reviewer raises an interesting point about comparing different datastore sources. We ran an additional experiment using datastores constructed from DCLM-baseline and FineWeb-Edu (which have been shown to perform better than Pile and Dolma). As Table R2 in the general reply shows, MassiveDS achieves comparable performance.
>
> ---
>
> **Weakness 3.** The claim that indexing the entire datastore initially reduces computational overhead is debatable. This aspect could be alleviated significantly by alternative indexing strategies [3, 4].
>
> **A3.** The reviewer is correct that embedding should be done once to optimize efficiency; this is also the core of our pipeline. However, there are additional challenges: e.g., the retrieval process is I/O bound and often takes significant time to load passages from a large document pool. Also, using existing software like FAISS requires building separate indices for different sets of dense embeddings, significantly increasing storage demands for various configurations. Our open-source pipeline provides an efficient, ready-to-use solution that overcomes these challenges and that can be used as a foundation for future research. We appreciate the reviewer’s point and will expand our discussion of this in the paper.
>
> The alternative indexing strategies cited by the reviewer do not fully overcome the challenges: [3] speeds up the search process but doesn't lower the cost of index construction; [4] improves datastore construction efficiency, but needs additional training and doesn't avoid redundant storage for different configurations. We will discuss them in our next revision.
>
> ---
>
> **Weakness 4 and Q3.** The overlap between the datastore and the downstream evaluation dataset raises concerns about data leakage.
>
> **A4.** We agree that data leakage is a potential concern. In Appendix B.1, we describe the strategies we used to avoid it. Specifically, we applied a strict data decontamination method to remove documents that overlap with the evaluation samples, which is stricter than common practice such as Dolma and RedPajama. In Figure 2(b), we also assessed perplexity (PPL) scaling on M2D2, which is not part of MassiveDS, and found scaling curves similar to those using decontaminated RPJ.
>
> ---
>
> **Weakness 5.** The author should reevaluate the claim with additional models.
>
> **A5.** We thank the reviewer for suggesting this. We have added more results with different LMs in Figure R1 in our general reply and will include them in our paper.
>
> ---
>
> **Q1.** How few-shot exemplars are selected for QA tasks?
>
> **A6.** Following the popular evaluation repository LM-Evaluation-Harness, we use few-shot examples randomly sampled from the development set, which do not overlap with test samples.
>
> ---
>
> **Q2.** How is the language model configuration determined? Is there any inherent randomness?
>
> **A7.** We follow LM-Evaluation-Harness, which uses greedy generation for TriviaQA and NQ, and log-likelihood for MMLU and MedQA. Therefore, there is no randomness in the inference. We’ll clarify this in the paper.
>
> ---
>
> **Q3.** (see Weakness 4)
>
> ---
>
> **Q4.** How do you substantiate that the contribution of MASSIVEDS is significant and not merely an assembly of various data segments?
>
> **A8.** Our primary contribution is enabling the scientific study of >1T-token RAG scaling on an academic budget, which extends beyond merely assembling the datastore and differs from previous systems that aimed to enhance search speed. Our identification of the commutability of different operations and the strategic ordering of these operations are crucial for facilitating accessible datastore scaling studies. Beyond this, we highlight our other contributions in presenting the first datastore scaling trends and analyses on various datastore design factors.
>
> We thank the reviewer for their valuable suggestions and we’ll update our draft accordingly.

---

> > ### Comment · Reviewer_AKfc · 2024-08-08
> > **Response to Authors**
> >
> > Dear Authors,
> >
> > Thank you for your detailed response. I appreciate the efforts to address my initial concerns, and I have accordingly updated my review score. However, I still have some follow-up questions:
> >
> > 1. What are the critical factors to consider when expanding a datastore with diverse data sources to ensure performance improvements in RAG?
> >
> > 2. Regarding the application of MASSIVEDS to other complex tasks, would you recommend maintaining the same datastore while adapting the retriever and language model to be more task-specific?
> >
> > 3. Could additional experiments be conducted on TriviaQA, using a merged corpus such as <MedQA corpus + TriviaQA corpus>, to assess whether such integration enhances performance?

---

> > > ### Author Response · Authors · 2024-08-11
> > > **Response to the follow-up questions**
> > >
> > > We thank the reviewer for raising the score, which encourages us. Here are some discussions on the follow-up questions:
> > >
> > > **Q1.** What are the critical factors to consider when expanding a datastore with diverse data sources to ensure performance improvements in RAG?
> > >
> > > **A1.** This is a great question. In Section 5.2, we  visualize the source distribution of retrieved results in Figure 3 and compare the performance of the multi-domain MassiveDS and single-domain datastores (Table 3). Our results show that the retriever tends to retrieve from relevant sources, and having more OOD data in the datastore does not have a negative impact on performance. Based on these observations, we hypothesize that having useful information in the datastore is the most critical factor. Therefore, our rule of thumb for data source selection is to **maximize the chance that there exists data in the datastore that can provide helpful information for future queries**. In other words, data sources that could potentially contain useful information for future queries are all desirable for better performance.
> > >
> > > Based on the above principle, we manually selected two types of data sources:
> > > The first type is general web data, such as data from CommonCrawl, which is a large and diverse data source that potentially covers various topics that a user may query at inference time.
> > > The second type is domain-specific data, which is selected based on our prior knowledge of the evaluation domain. For example, we intentionally added more scientific data, such as pes2o scientific papers, math datasets, and biomedical data, to the datastore because we want the RAG system to perform better on scientific benchmarks such as MMLU and MedQA.
> > >
> > > In summary, our data source selection was done empirically and we believe that exploring automatic data selection methods for datastore construction is a promising future direction. For instance, one potential follow-up work could involve training classifiers for source-level or document-level data selection to reduce the size of the datastore while maintaining its effectiveness for targeted tasks. We will include these discussions in the updated paper.
> > >
> > >
> > > **Q2.** Regarding the application of MASSIVEDS to other complex tasks, would you recommend maintaining the same datastore while adapting the retriever and language model to be more task-specific?
> > >
> > > **A2.** As discussed in A1, when adapting to new tasks, we recommend optimizing the datastore composition to include more data sources that are potentially helpful for these new tasks. Prior knowledge about the task distribution can help guide targeted data selection. For example, you may want to include more code data when building a datastore for coding tasks, such as HumanEval.
> > >
> > > **Q3.** Could additional experiments be conducted on TriviaQA, using a merged corpus such as <MedQA corpus + TriviaQA corpus>, to assess whether such integration enhances performance?
> > >
> > > **A3..** We would like to clarify that including more data in the datastore is likely, but not guaranteed, to improve performance. However, you can still try including everything in a datastore because it is robust to OOD data if there is no storage or computational constraints for the datastore size. To illustrate this, we conducted additional experiments where we compared the performance of different datastore combinations using the setup suggested by the reviewer. The results are shown in the below table.
> > >
> > > | Datastore     | LM-only | Wikipedia | Wikipedia + C4 | Wikipedia+PubMed | MassiveDS |
> > > |---------|------|---|-------|---------|------|
> > > | TQA             |    64.1   | 72.6      | 75.8           | 72.6              | 77.0      |
> > >
> > > As shown in the table, adding a new data source that is a hybrid of OOD data and potentially helpful data, such as C4, can further improve the performance of the in-domain datastore. Meanwhile, including a data source that only contains OOD data, such as PubMed, does not improve or decrease the TQA performance. However, we note that having PubMed in the datastore is helpful for biomedical tasks. Therefore, we recommend increasing the diversity of the datastore to enhance the performance on both single or multiple tasks. We will clarify that the data sources should still be carefully chosen with a consideration on what sources would be potentially helpful to the targeted tasks.
> > >
> > > We thank the reviewer for the insightful discussion. We are happy to follow up if the reviewer has any other questions!

---

> > > > ### Comment · Reviewer_AKfc · 2024-08-11
> > > > **Response to Authors**
> > > >
> > > > Dear Authors,
> > > >
> > > > Thank you for addressing my questions. A1 has resolved most of my concerns regarding the generalization of the scaling trend, and I recommend incorporating it into the main paper, as it is well justified. Regarding W3, in my initial reading, I assumed the computational overhead was related to GPU resources. It would be helpful to clarify this point as discussed in previous A3. Additionally, it would be beneficial to discuss whether these computational challenges are theoretically inherent to the method or if they stem from the choice of index type and third-party libraries. I have made the necessary changes to the score based on our discussion.

---

> > > > > ### Author Response · Authors · 2024-08-12
> > > > >
> > > > > We thank the reviewer for their insightful suggestions, which are very valuable in helping us further improve the quality of our paper. We also appreciate the reviewer's raised score. We will definitely include these discussions in our next revised version. Thank you for taking the time to review our manuscript!

---

### Official Review · Reviewer_vyTD · 2024-07-10

**Soundness:** 3
**Presentation:** 4
**Contribution:** 2
**Rating:** 6
**Confidence:** 4

**Summary:**

This paper studies the effect of scaling datastores for retrieval-based language models. A trillion-token datastore, MassiveDS, is constructed and then filtered to remove contaminated and duplicate documents. A distributed pipeline is proposed to index and retrieve from MassiveDS with a modest compute budget. Evaluation on language modeling tasks and downstream QA tasks shows that scaling the datastore brings clear improvements, and careful datastore filtering is crucial for model performance.

**Strengths:**

1. This paper studies an important research question: the scaling of datastores for retrieval-based LMs.

2. This paper open-sources the 1.4T token datastore for retrieval-based LMs, which consists of diverse domains and is carefully filtered. This datastore could be a valuable resource for future retrieval-based LM studies.

3. Evaluation on multiple tasks demonstrates the importance of datastore scaling and highlights some interesting questions, such as how to improve retrieval performance and how to filter datastores.

**Weaknesses:**

1. The claim of proposing the 'largest' datastore could be reconsidered. For LM pre-training, there are larger datastores like RedPajama-v2 with 30T tokens, and for IR, there are corpora like ClueWeb22 with 16T tokens. The proposed MassiveDS is mainly sourced from the RedPajama, making it similar to other existing collections.

2. The proposed distributed index and retrieval pipeline is interesting but seems inflexible. How does it compare with existing search engine frameworks like ElasticSearch and Weaviate?

3. This paper lacks an analysis of retriever. Contriever-msmarco is used, whose training data may differ from MedQA and MMLU. Considering the limited generalizability of dense retrievers (as shown by the BEIR paper), stronger retrievers like those in MTEB could be used.

**Questions:**

1. What is the recall rate of the retriever for different datastore sizes?

**Limitations:**

The limitations section is included and adequately discussed.

---

> ### Author Rebuttal · Authors · 2024-08-07
>
> We thank the reviewers for acknowledging the importance of our research question and the value of our findings and open-sourced resources! We would like to address the reviewer’s concerns below:
>
> **Weakness 1.** The claim of proposing the 'largest' datastore could be reconsidered as there are other pretraining corpus that exceed even 10T tokens.
>
> **A1.** We would like to clarify that a datastore is different from a pretraining corpus: by “datastore”, we refer to an index that is ready for retrieval [1]; while a pretraining corpus only contains the raw data. Building a datastore on an existing pretraining corpus requires further data cleaning, data chunking, embedding, and indexing over the raw data, which is a non-trivial effort for the retrieval community. It has been challenging to conduct datastore scaling research at this scale before and existing open-sourced datastores are much smaller than MassiveDS, as shown in Table 1. Althogh there is pretraining corpus that is larger than our datastore in terms of raw text, we are the first to make this attempt to construct and open-source a datastore with over 1 trillion tokens.
>
> [1] Asai, Akari, et al. "Reliable, adaptable, and attributable language models with retrieval."
>
> ---
>
> **Weakness 2.** The proposed distributed index and retrieval pipeline is interesting but seems inflexible. How does it compare with existing search engine frameworks like ElasticSearch and Weaviate?
>
> **A2.** We would like to note that our contribution is orthogonal to these existing frameworks: prior work, such as ElasticSearch and Weaviate, focuses on efficiency at inference, while our work focuses on the scientific evaluation of retrieval performance which requires not efficient serving, but efficient experimentation of various factors of the datastore. Even with the state-of-the-art nearest neighbor search used in production, studying the effect of various factors of the datastore like the size, quality filters, and decontamination methods remains expensive. This is because such experiments require rebuilding search indices for every combination of the factors, orthogonal to what search algorithm is being used. Our study focuses on removing the needs for such repetitive rebuilding of indices. The outcome is the comprehensive study of the impact of various datastore factors, as we demonstrated as part of the results, which is the novel part of our work.
>
> ---
>
> **Weakness 3.** This paper lacks an analysis of retriever. Contriever-msmarco is used, whose training data may differ from MedQA and MMLU. Considering the limited generalizability of dense retrievers (as shown by the BEIR paper), stronger retrievers like those in MTEB could be used.
>
> **A3.** We agree with the reviewer that we indeed only investigated one base retriever due to being limited by academic computational resources, and we studied the effect of improved retrieval by applying a cross-encoder reranker which is more computationally efficient in the paper. In the table below, we run the evaluation using 2 more base retrievers of similar sizes that perform better than Contriever on the MTEB benchmark using 10% randomly sampled MassiveDS. Our results show that, interestingly, the performance of the base retrievers of similar sizes on our general web data does not necessarily align with the ranking on MTEB. We hypothesize that this low correlation is because the domain compositions of MassiveDS and MTEB are different. Due to limited computational resources, we defer the study with larger retrievers, such as GRIT-7B, to future work. But we note that such larger embedding models are often prohibitively expensive to scale up to a trillion-token datastore (Wang et al., 2024).
> | Name       | Retriever Type | Size  | Perplexity ↓ | Natural Questions ↑ | MMLU ↑ |
> |------------|----------------|-------|--------------|----------------------|--------|
> | Contriever | dense          | 177M  | 4.2210       | 0.3321               | 0.4922 |
> | DRAGON     | dense          | 110M  | 4.2373       | **0.3399**           | 0.4875 |
> | GTR-Base   | dense          | 110M  | **4.2146**   | 0.3080               | **0.4934** |
>
> ---
>
> **Question 1.** What is the recall rate of the retriever for different datastore sizes?
>
> **A4.** In our evaluation setup, both upstream language modeling and downstream evaluation sets do not provide gold documents for the questions. Therefore, we can only report the end-to-end performance on these tasks.

---

### Official Review · Reviewer_omwP · 2024-07-11

**Soundness:** 3
**Presentation:** 4
**Contribution:** 3
**Rating:** 7
**Confidence:** 3

**Summary:**

This paper studies the impact of scaling the datastore (retrieval dataset) on retrieval-based language models.

The contributions are:
- MASSIVEDS a 1.4 trillion-token datastore for retrieval-based LMs that will be made open-source.
- A pipeline to study the impact of the datastore scaling on the language models at inference time
- An analysis of the impact of scaling and other pre-processing steps (decontamination, deduplication, and quality filtering) on the language models at inference time

**Strengths:**

1) OPEN SOURCE

Both the models and data used in the work are open source and when the code is made available, it will facilitate future work and additional research on retrieval-based LM at scale.

2) WRITING/PRESENTATION

The paper is easy to read, the contributions and findings are explicitly stated, and the tables and figures are easy to read.

3) IMPACT

Given the importance of retrieval augmented generation for making LM more trustworthy and adapted to a specific domain/environment, releasing a large-scale dataset and pipeline for retrieval-based LMs can facilitate future research in that domain.

**Weaknesses:**

INFERENCE TIME ANALYSIS

This work does not study the impact of the scaling of the datastore on inference time. Since scaling the data store can make retrieval slower and therefore, make the generation slower, this is a crucial aspect to study, that was not addressed in this work.

**Questions:**

1) CODE NOT AVAILABLE?

Unless there is an oversight on my part, I could not find the implementation (no link to anonymous GitHub and no supplementary material) and I could not find the code attached to the paper. Although the authors indicated that the code and data will be made available, having an anonymous GitHub repo with the code already available would improve the strength of this submission.

2) RETRIEVAL AUGMENTED GENERATION?

Since the LMs used in the experiment are all generative language models, why is the term Retrieval Augmented Generation not used to describe this work? ]

3) DATA DECONTAMINATION/DEDUPLICATION

Can more details be provided about data decontamination/deduplication? In practice, what is the difference between the two? Since data decontamination filters out "documents with 80+% 13-gram Jaccard similarity or 32-gram longest sequence overlap", how is deduplication made? is it only on documents matching exactly at the string level?

4) FIGURE 4 (small detail):

Is there a reason why the baselines LM-only on the subplots (d), (e), and (f) are not extended with a dashed line?

**Limitations:**

The authors discuss several limitations of their work:
- Focus on a particular class of retrieval-based LMs (RIC-LMs),
- Do not study very large LMs (only 1B and 7B parameters)
- Only consider dense retrievers.
- Only considers QA tasks

There is only one limitation to this work that was not mentioned by the authors: the lack of analysis of the impact on inference time (see weaknesses).

---

> ### Author Rebuttal · Authors · 2024-08-07
>
> We thank the reviewer for highlighting our open-source contribution and acknowledge the potential impact on the community! We would like to address the reviewer’s concerns below:
>
> **Weakness 1.** This work does not study the impact of the scaling of the datastore on inference time.
>
> **A1.** Our main focus is to see the effect of dataset scaling on upstream and downstream model performance. While we agree that a larger datastore can introduce additional latency, there's active research on improving the speed of nearest neighbor search; we believe our contributions are orthogonal. In addition, we show a small LM with retrieval augmentation could match or even outperform larger LM-only models, as supplemented in Figure R1 in the general reply, which indicates a potential reduction of inference time by using a small LM augmented with MassiveDS. Given the complexity of inference speed optimization and its loose relationship with our focus, we defer the study of inference-time efficiency-performance tradeoff to future work.
>
> ---
>
> **Question 1.** The codes are not available by the time of submission.
>
> **A2.** We thank the reviewer for reminding us to provide anonymous codes with the submission. We uploaded our code in an anonymous github repository and sent it to the AC according to the rebuttal rules. We hope the reviewer could easily get access to the code.
>
> ---
>
> **Question 2.** Why don’t the authors use the term “retrieval augmented generation (RAG)”?
>
> **A3.** We define retrieval-based LMs as a general family of LMs that leverage large-scale text data at inference time [1], and RAG [2] is one such model. We believe that our finding is generally applicable to other types of retrieval-based LMs such as kNN-LM style models, and therefore decided to use the term. More specifically, we focus on retrieval-in-context LMs as mentioned in Section 2, which is often used interchangeably with RAG [3] but more specifically describes off-the-shelf LMs that use retrieved context at inference time.
>
> [1] ACL 2023 Tutorial: Retrieval-based Language Models and Applications https://acl2023-retrieval-lm.github.io/
>
> [2] Lewis, Patrick, et al. "Retrieval-augmented generation for knowledge-intensive nlp tasks."
>
> [3] Min, Sewon, et al. "Silo language models: Isolating legal risk in a nonparametric datastore."
>
> ---
>
> **Question 3.** Can more details be provided about data decontamination/deduplication?
>
> **A4.** The motivation of deduplication for a datastore is to remove documents that are exactly or approximately the same such that the retrieved top-k documents won’t contain repetitive information. To achieve this, a similarity score is computed between every two documents in the datastore, and one document is removed from every duplicate pair. While decontamination aims at removing the same or near-duplicate documents in the datastore that directly contain the information about the test samples. The motivation is to avoid having test samples directly in the datastore such that the model can achieve high evaluation performance by cheating. The test sample is compared against every document in the datastore, and the documents with high similarity scores are removed from the datastore for this certain task.
> In our paper, the 13-gram Jaccard similarity score is one metric we use to measure the overlapping rate between two documents (in deduplication, it’s 2 documents from the datastore; in decontamination, one document is the test sample and another document is from the datastore). We set the threshold to 80% for both deduplication and decontamination that we remove a document if it has a similarity score that is higher than 80% with another document. Since decontamination plays an important role in evaluation, we combined the 13-gram Jaccard similarity metric with another metric for decontamination to make our datastore less likely to be contaminated. More details can be found in Appendix B.1.
>
> ---
>
> **Question 4.** Is there a reason why the baselines LM-only on the subplots (d), (e), and (f) are not extended with a dashed line?
>
> **A5.** We thank the reviewer for pointing out this detail! We forgot to add a dashed line for these 3 subfigures and we will add them back in the next revised version!

---

> > ### Comment · Reviewer_omwP · 2024-08-12
> >
> > Thank the authors for further explanations. I do not have other concerns and will keep my original positive rating.

---

### Official Review · Reviewer_TsAT · 2024-07-17

**Soundness:** 4
**Presentation:** 4
**Contribution:** 3
**Rating:** 7
**Confidence:** 3

**Summary:**

The paper introduces MASSIVEDS, the largest and most diverse open-sourced datastore for retrieval-based language models, containing 1.4 trillion tokens. The authors design a MASSIVEDS pipeline to efficiently explore the impact of different datastore features by reordering the datastore construction operations so that the most expensive operations, such as indexing and retrieval, are only run once. Extensive experiments demonstrate that model performance improves as the datastore size increases.

**Strengths:**

(1) To my knowledge, this is the first work studying scaling laws regarding datastore size. The author introduces the largest and most diverse open-sourced datastore for retrieval-based language models, containing 1.4 trillion tokens. This large-scale datastore can serve as a promising testbed for developing new retrieval-based LMs.

(2) The author promises to release not only the data but also the code to reproduce the experiments. This could greatly facilitate research efforts within the community.

(3) The author proposes a simple yet effective data processing pipeline that enables efficient investigation of the impact of different datastore features.

(4) Extensive experiments show that the performance of retrieval-based LMs benefits from scaling datastore sizes. The evaluation includes both language modeling and downstream question answering. The results on MMLU and MedQA also highlight the need for developing retrieval-augmented LMs that can excel in tasks requiring reasoning abilities.

(5) Some ablation studies provide interesting and insightful findings, such as data deduplication being a crucial factor for enhancing language model performance when the datastore size is extremely large.

Overall, I believe this is a good paper studying an important problem supported by solid experiments.

**Weaknesses:**

Overall, I do not find any significant weaknesses in the paper. The only minor issues I noticed are:

(1) The retriever and reranker used in this paper may be somewhat outdated. Recent models on the MTEB leaderboard might offer better performance. The benefits of scaling datastore sizes could be more significant with a more powerful retriever. However, this is a very minor point. Considering the computational cost of indexing all the data, trying other retrievers is not needed.

(2) Typo: L223 "Datascore" should be "Datastore".

**Questions:**

Please see my comments in Weakness.

**Limitations:**

The limitations are properly stated in the Conclusion and Limitations section.

---

> ### Author Rebuttal · Authors · 2024-08-07
>
> We thank the reviewer for acknowledging the importance of our study and the solidness of our work! We would like to address the only concern by the reviewer on the choice of retriever below.
>
> **Q1.** The retriever and reranker used in this paper may be somewhat outdated. Recent models on the MTEB leaderboard might offer better performance. The benefits of scaling datastore sizes could be more significant with a more powerful retriever. However, this is a very minor point. Considering the computational cost of indexing all the data, trying other retrievers is not needed.
>
> **A1.** We thank the reviewer for understanding our computational constraints, and we agree with the reviewer that the base retriever we used in this paper isn’t new. We chose Contriever-MSMACRO because it was used in many previous works cited in L178-179. We agree that benefits of scaling datastore sizes could be more significant with a more powerful retriever, which is consistent with our conclusion in Section 6.2 that improved retrieval could enhance scaling trends. In the table below, we tried 2 more base retrievers that outperform Contriever on the MTEB leaderboard using 10% randomly sampled MassiveDS. Our results show that, interestingly, the performance of the base retrievers of similar sizes on our general web data does not necessarily align with the ranking on MTEB. We hypothesize that this low correlation is because the domain compositions of MassiveDS and MTEB are different. Due to limited computational resources, we defer the study with larger retrievers, such as GRIT-7B, to future work. But we note that such larger embedding models are often prohibitively expensive to scale up to a trillion-token datastore (Wang et al., 2024).
> | Name       | Retriever Type | Size  | Perplexity ↓ | Natural Questions ↑ | MMLU ↑ |
> |------------|----------------|-------|--------------|----------------------|--------|
> | Contriever | dense          | 177M  | 4.2210       | 0.3321               | 0.4922 |
> | DRAGON     | dense          | 110M  | 4.2373       | **0.3399**           | 0.4875 |
> | GTR-Base   | dense          | 110M  | **4.2146**   | 0.3080               | **0.4934** |
>
> ---
>
> **Q2.** Typo: L223 "Datascore" should be "Datastore".
>
> **A2.** We thank the reviewer for pointing out the typo for us. We will fix it in our next version. We are happy to address more questions from the reviewer during the rebuttal period.

---

### Official Review · Reviewer_cEKQ · 2024-07-28

**Soundness:** 4
**Presentation:** 4
**Contribution:** 2
**Rating:** 7
**Confidence:** 4

**Summary:**

The paper "Scaling Retrieval-Based Language Models with a Trillion-Token Datastore" explores a new dimension of scaling language models (LMs) by considering the amount of data used during inference. The study focuses on retrieval-based LMs, which access a large external datastore during inference, and examines the impact of scaling the datastore. The authors introduce MASSIVEDS, a 1.4 trillion-token datastore, and an efficient pipeline to study various datastore features such as size, data filters, and decontamination strategies. The experiments reveal that datastore scaling follows a log-linear trend and significantly improves performance on various tasks. The authors also highlight the importance of aggressive decontamination, data deduplication, and improved retrieval techniques for better scalability. If accepted, the dataset MASSIVEDS will be the largest and also most diverse open-sourced datastore which will benefit the research community.

**Strengths:**

- **Originality**: The paper tackles the important question of how scaling retrieval-based LMs by the size of the inference datastore impacts upstream and downstream task performance which is novel and adds a new dimension to LM scaling laws.
- **Quality**: The research is rigorous, with well-designed experiments and robust methodologies.
- **Clarity**: The paper is well-organized, and the key contributions are clearly articulated.
- **Significance**: The open-sourced resources will facilitate further research in this direction.
- **Experiment Design**: The design of the experimental set-up is novel and theoretically grounded which will be interest to the research community on how to efficiently experiment with multiple dataset configurations

**Weaknesses:**

The following are some of the limitations of the paper (some of these have already been highlighted by the authors):
- **Task Diversity**: The evaluation is primarily focused on QA and knowledge-intensive tasks where one can expect the retrieval based LMs to work well. Inclusion of more diverse tasks, such as commonsense reasoning or open-ended text generation, would provide a more comprehensive assessment of the datastore's scalability.
- **Limited Scope of Analysis**: This is a generic comment about the overall analysis results in the paper. For each of the analysis dimension, the authors have scratched the surface by experimenting with a basic technique which I am afraid makes the results limited in their scope and generalizability. Here are a few examples to illustrate the point - a) quality filtering: they explore basic filtering from DOLMA but leave the higher-quality filters for future-work b) Decontamination: 13-gram models for decontamination and leave out some techniques such as Min-K% Prob which are known to be stronger baselines c) Retrieval Model: The paper uses a specific retrieval model (CONTRIEVER-MSMARCO) without exploring alternative models. This might be due to the fact that a major portion of the paper was focused designing the scalable experimental set-up for this analysis.

**Questions:**

1. Previous studies (https://arxiv.org/html/2307.07164v2) have shown that lack of diversity adversely impacts the results. What was the diversity of results retrieved by the underlying retrieval?

**Limitations:**

The authors adequately address the limitations of their work, such as focusing on a specific class of retrieval-based LMs, limited model sizes, and dense retrievers. They also acknowledge the need for further exploration of different retrieval architectures and a wider range of evaluation tasks.

---

> ### Author Rebuttal · Authors · 2024-08-07
>
> We thank the reviewer for acknowledging the novelty and solidness of our work! We would like to address the concerns and questions below:
>
> **Weakness 1.** The evaluation is primarily focused on QA and knowledge-intensive tasks where one can expect the retrieval based LMs to work well. Inclusion of more diverse tasks, such as commonsense reasoning or open-ended text generation, would provide a more comprehensive assessment of the datastore's scalability.
>
> **A1.** We agree that more diverse tasks would make our study more comprehensive. Our study takes a step in this direction as it is the first analysis, to the best of our knowledge, of datascore scaling on downstream tasks. Previous studies (RETRO, Borgeaud, et al.) examine retrieval-based language models with datastores exceeding 1 trillion tokens have focused solely on language modeling, and it remains unclear how datastore scaling and different datastore factors impact the performance of downstream tasks.
> In addition to knowledge-intensive tasks, we also include tasks that involve reasoning on top of knowledge. In particular, we included MMLU, which is widely recognized as a standard metric for assessing the performance of modern LMs, in our evaluation. Given our resource constraints, it was challenging to cover more downstream tasks, as we discussed in our limitation sections. We will emphasize this point in the next revision, and we hope that by open-sourcing our datastore and pipeline, we will enable others to build upon our work and evaluate large-scale retrieval-based models on a more diverse set of downstream tasks.
>
> ---
>
> **Weakness 2.** For each of the analysis dimensions, the techniques are basic: E.g., a) quality filtering: basic filtering from DOLMA but leave the higher-quality filters b) Decontamination: 13-gram models for decontamination and leave out some techniques such as Min-K% Prob c) Retrieval Model. This might be due to the fact that a major portion of the paper was focused designing the scalable experimental set-up for this analysis.
>
> **A2.** We thank the reviewer for acknowledging that our main focus is on the design of scalable experimental setup, as it is challenging to cover all possible datastore configurations given our computational constraints. Nevertheless, our analyses have covered several key aspects that people care about when designing the datastores, and our findings are inspiring.
>
> Here are some discussions on the points mentioned by the reviewer:
>
> (1) We agree with the reviewers that the DOLMA filters are basic and that higher-quality filters, such as fasttext filtering [1], are not covered in our study. We focus on standard filters that have been applied to both traditional and modern pretraining corpora such as RedPajama, C4, Dolma, etc. We believe these analyses are interesting to a broad community. In addition, we show that it is easy to test the effect of different quality filters using our pipeline and defer the study on more recent filters to future works.
>
> (2) For data decontamination, we considered 13-gram Jaccard similarity-based decontamination because it has been widely adopted by many works, such as GPT-3, RETRO, Dolma, etc.  Beside this, we also applied another decontamination method, longest-string decontamination, which enables us to experiment with different strictness levels of decontamination, as shown in Figure 4. By tuning the hyper-parameters of the decontamination methods, we show datastore scaling can benefit the model’s performance across different decontamination levels (Figure 4), so our conclusions hold regardless of the strictness of the decontamination method. Separately, Min-K% prob is designed for detecting if a given LM has been trained on certain data, and isn’t intended to be a data decontamination method.
>
> (3) We indeed only used a single base retriever model in the submission, so we add a new ablation on the base retriever using 10% of MassiveDS and supplement the results in the table below. We chose DRAGON and GTR-Base because they rank higher than Contriever on the MTEB benchmark and are similar in size. Interestingly, the results indicate that these base retrievers exhibit similar performance on MassiveDS. We hypothesize that this is due to the differing domain compositions of MassiveDS compared to the datasets used in MTEB.
>
> | Name       | Perplexity ↓ | Natural Questions ↑ | MMLU ↑ |
> |------------|--------------|----------------------|--------|
> | Contriever | 4.2210       | 0.3321               | 0.4922 |
> | DRAGON     | 4.2373       | 0.3399           | 0.4875 |
> | GTR-Base   | 4.2146   | 0.3080               | 0.4934 |
>
> Due to limited compute, we leave it to future work to explore the performance with a larger and more capable retriever such as GRIT-7B. But we note that such larger embedding models are often prohibitively expensive to scale up to a trillion-token datastore (Wang et al., 2024). However, we are optimistic that significantly improving the retriever will lead to better scaling trends. This expectation is supported by the evidence in Figure 5, which shows that enhancing retrieval quality with a cross-encoder reranker improves performance.
>
> [1] Li, Jeffrey, et al. "DataComp-LM: In search of the next generation of training sets for language models."
>
> ---
>
> **Q3.** Previous studies (https://arxiv.org/html/2307.07164v2) have shown that lack of diversity adversely impacts the results. What was the diversity of results retrieved by the underlying retrieval?
>
> **A3.** As shown in Figure 3, the retriever retrieves from diverse sources in MassiveDS, and it tends to retrieve from relevant domains more frequently than other domains. Our results also indicate that increasing diversity helps improve the performance—MassiveDS outperforms single-domain datastores on the evaluated tasks, as shown in Table 3. We will discuss this in our next version.

---

> > ### Comment · Reviewer_cEKQ · 2024-08-11
> > **Response to Rebuttal from Authors**
> >
> > Dear Authors, Thanks for responding to my comments and also including the supplementary results. The response and the proposed revisions to the next version adequately addresses the concerns I had related to some of the weakness of this work. After going through their responses, I have no other major concerns and due to this I am revising my rating.

---

### Official Review · Reviewer_F6AB · 2024-07-29

**Soundness:** 2
**Presentation:** 3
**Contribution:** 3
**Rating:** 5
**Confidence:** 4

**Summary:**

The paper studies the effects of scaling retrieval datastore in retrieval augmented language models. The authors present the impacts of various design choices such as data size and data selection. A testbed dataset MassiveDS is also introduced.

**Strengths:**

- The authors present a substantially larger retrieval datastore for retrieval augmented LMs.
- Extensive experiments have been conducted to study how datastore designs/properties can impact RAG results.
- Detailed analysis has been conducted on several eval datasets and models.

**Weaknesses:**

- Large-scale embedding-based search is not a new topic but has been studied for years. Arguably it is a mature technology ready in production [1, 2].  The authors have made a rather arbitrary design in the index design. Information like hardware latency and memory/CPU footprint were not reported; comparisons with popular techniques like DiskANN [2] are not presented. This could be confusing and/or misleading to people reading the paper: should they adopt the search pipeline introduced in the paper, or just use a more developed technology? Also, this makes it difficult to assess the contribution of the entire paper.
- While the paper shows a general improving trend in RAG tasks and with relatively capable LMs, the models in the paper still seem to underperform the popular 2-year-old RAG model ATLAS [3] by a significant margin. Specifically, the design of the RAG system used in the paper can be somewhat basic. How much the results transfer to stronger system/system designs remains relatively unclear.


[1] Huang, J., Sharma, A., Sun, S., Xia, L., Zhang, D., Pronin, P., Padmanabhan, J., Ottaviano, G., & Yang, L. (2020). Embedding-based Retrieval in Facebook Search. Proceedings of the 26th ACM SIGKDD International Conference on Knowledge Discovery & Data Mining.

[2] Subramanya, S.J., Devvrit, Kadekodi, R., Krishaswamy, R., & Simhadri, H.V. (2019). DiskANN : Fast Accurate Billion-point Nearest Neighbor Search on a Single Node.

[3] Izacard, G., Lewis, P., Lomeli, M., Hosseini, L., Petroni, F., Schick, T., Yu, J.A., Joulin, A., Riedel, S., & Grave, E. (2022). Few-shot Learning with Retrieval Augmented Language Models. ArXiv, abs/2208.03299.

**Questions:**

- It was discussed in the paper that MASSIVEDS can outperform in-domain datastores on two eval sets. I wonder how this happens. Are there pieces of information easier to retrieve and/or read from out-of-domain? If so, what forms can this information take to be easier to read and retrieve than the original information?
 - The authors claim to observe an empirical log-linear scaling trend in the RAG system. It would be interesting to see some discussion on why this is the case and the implications of this. How does this happen? Does this mean retrieval datastore scaling has an extremely diminishing gain?
- The authors use token counts extensively across the paper. While this is common in describing pre-training data, the actual retrieval units, i.e. passage/chunk is arguably a better measurement to describe the statistics of the retrieval task. Can the authors provide some statistics based on chunks/passages?

**Limitations:**

n.a.

---

> ### Author Rebuttal · Authors · 2024-08-07
>
> We thank the reviewer for pointing out the comprehensiveness of our work! We would like to address the concerns and questions below, and we will edite the paper accordingly.
>
> **Weakness 1.** Large-scale embedding-based search is not a new topic but has been studied for years. Arguably it is a mature technology ready in production [1, 2]. Information like hardware latency and memory/CPU footprint were not reported; comparisons with popular techniques like DiskANN [2] are not presented. This could be confusing and/or misleading to people reading the paper: should they adopt the search pipeline introduced in the paper, or just use a more developed technology? Also, this makes it difficult to assess the contribution of the entire paper.
>
> **A1.** The work listed by the reviewer focuses on the search efficiency at inference with a fixed datastore configuration, but we would like to clarify that this is orthogonal to the focus of the current paper. Our main goal is to conduct a thorough analysis of the effect of scaling the datastore for retrieval-based language models and the impact of various datastore factors, where the inference speed isn’t a key bottleneck—even with the state-of-the-art nearest neighbor search used in productions such as DiskANN, studying the effect of various factors of the datastore like the size, quality filters, and decontamination methods remains expensive. This is because such experiments require rebuilding search indices for every combination of the factors, unrelated to which search algorithm is being used. Therefore, we design a new pipeline for efficient experimentation of various factors of the datastore by removing the need for repetitive datastore construction and repetitive large-scale retrieval, which are orthogonal and can be combined with prior work on efficient search algorithms.
>
> ---
>
> **Weakness 2.** While the paper shows a general improving trend in RAG tasks and with relatively capable LMs, the models in the paper still seem to underperform the popular 2-year-old RAG model ATLAS [3] by a significant margin. Specifically, the design of the RAG system used in the paper can be somewhat basic. How much the results transfer to stronger system/system designs remains relatively unclear.
>
> **A2.** Despite active research on advancing RAG designs, there have been few scientific studies of the scaling properties of datastores with more than a few billion tokens. As an initial open-source study on datastore scaling, we picked the most basic but widely adopted RAG design instead of any particularly optimized system. In Figure 5, we presented an analysis of the impact of improved retrieval on the datastore scaling performance, which shows that adopting a more advanced retrieval technique, i.e., a reranker, can further improve the scaling performance, indicating a positive sign that our conclusions could potentially be extended to more recent RAG designs. Overall, we focus on conducting a comprehensive analysis of datastore scaling and sharing our thoughts on how future works could further improve the scaling curves, rather than achieving state-of-the-art scores on specific datasets with a particularly optimized RAG design.
>
> The purpose of Atlas is to finetune the retriever for best performance. Atlas finetunes the retriever and the LM to adapt to every specific task using the full task data (>10k) for their best-reported scores, while we evaluate the model in a training-free fashion by prepending only 5 examples in context. The purpose of our work is to disentangle the factors that affect performance and study them in detail. As such, Altas is orthogonal work. The insights from our work can be combined with Atlas, but here we ask more basic questions about RAG scaling.
>
> ---
>
> **Question 1.** It was discussed in the paper that MASSIVEDS can outperform in-domain datastores on two eval sets. I wonder how this happens. Are there pieces of information easier to retrieve and/or read from out-of-domain?
>
> **A3.** MassiveDS is a hybrid multi-domain datastore that includes both “in-domain datastores” as subsets and general web data to enhance its knowledge base. The key finding here is that the model is robust to other out-of-domain data in the same datastore and the retriever still preferentially retrieves from the right domain (Figure 3), so it eliminates the need to develop domain-specific datastores for each task, which is often a costly and complex process. Consequently, users can establish a single, general-purpose datastore that is effective across various tasks, even in cases where creating a task-specific in-domain datastore is unfeasible. This flexibility greatly simplifies data management and expands the utility of MassiveDS in diverse applications.
>
> ---
> **Question 2.** The authors claim to observe an empirical log-linear scaling trend in the RAG system. It would be interesting to see some discussion on why this is the case and the implications of this. How does this happen? Does this mean retrieval datastore scaling has an extremely diminishing gain?
>
> **A4.** Many previous scaling law studies have shown significant gains by scaling the pretraining data and the model size, and they all present performance gains with the x-axis (the number of tokens for pretraining or the number of parameters) in the log scale [1,2]. We find similar performance gains by scaling the datastore to the gains by scaling pretraining data. Therefore, our results indicate that the datastore could be another dimension to scale in addition to the pretraining data and the model size.
>
> [1] Gadre, Samir Yitzhak, et al. "Language models scale reliably with over-training and on downstream tasks."
>
> [2] Kaplan, Jared, et al. "Scaling laws for neural language models."
>
> ---
> **Question 3.** Can the authors provide some statistics based on chunks/passages?
>
> **A5.** We have 4B passages in our datastore, which average around 360 tokens. We will report detailed statistics in the next version of the paper.

---

> ### Author Response · Authors · 2024-08-13
>
> Dear Reviewer,
>
> As the discussion deadline approaches, we want to ensure that all your concerns and questions have been fully addressed. To address the weaknesses listed in the original review, we have clarified our contributions and the detailed settings of our experiments. We will revise our paper accordingly.
>
> We deeply appreciate your insights and the time you have invested in reviewing our work. Your feedback is invaluable in refining our research and ensuring its quality and relevance. Please let us know if you have any further questions!

---

### Author Rebuttal · Authors · 2024-08-07

We appreciate the reviewers' strong support for the contributions of the paper and their insightful comments. This general response outlines how we have responded to their concerns and provides the requested supplementary results.

**Summary of common concerns and our corresponding response.**
* Reviewer F6AB and Reviewer TsAT asked to compare our proposed pipeline with existing retrieval systems for efficient search at inference time, such as DiskANN. We would like to clarify that our pipeline is optimized for a different goal and is orthogonal to the existing systems.
    * Our pipeline is designed to facilitate accessible datastore scaling studies, where the key challenge is how to efficiently study datastores of different configurations, such as datastore sizes, quality filters, data decontamination methods, etc. While prior work such as DiskANN focuses on search efficiency at inference time given a fixed configuration, we believe these are complementary to our work. Specifically, even with the state-of-the-art nearest neighbor search used in production, studying the effect of various factors of the datastore remains expensive. This is because such experiments require rebuilding search indexes for every combination of the factors, orthogonal to what search algorithm is being used. Our work focuses on removing the need for such repetitive rebuilding of indexes. This allows us to conduct  a comprehensive study of the impact of various datastore factors, which is a novel contribution of our work.

* Reviewer TsAT and Reviewer AKfc asked about comparisons of MassiveDS with other datasets such as Dolma. We would like to clarify that the definition of a datastore is different from a pretraining dataset.
    * By “datastore”, we refer to an index that is ready for retrieval; a pretraining corpus only contains raw data. Building a datastore on an existing pretraining corpus requires additional data chunking, embedding, and indexing over the raw data, which is a non-trivial effort. It has been challenging to conduct datastore scaling research at this scale before and the existing open-sourced datastores are much smaller than MassiveDS, as shown in Table 1 of the submission. Prior larger scale pre-training data is only available in the format of raw text, while we open source both the raw text as well as the resulting embedding and index to search from the entire 1.4 trillion corpora. In addition, we open source the codebase to use our released index for future study.
    * Inspired by the reviewer, we further compare the performance of datastores built with different sources in Table R2 below, which hasn’t been examined by any prior work. See the response below for more details.

* Requested supplementary results.

    * As requested by Reviewer omwP and Reviewer AKfc, we supplement more scaling results with more language models, such as Llama3, Olmo, and Pythia models, in Figure R1. The new results show our conclusions hold across different LMs.

    * As requested by Reviewer cEKQ, Reviewer TsAT, Reviewer vyTD, and Reviewer AKfc, we supplement the evaluation results using 3 different base retrievers in Table R1. The results indicate that retrievers of similar sizes have comparable performance on general-web data and that Contriever-MSMACRO is a reasonable pick for our main experiments.

    * As requested by Reviewer AKfc, we compare the performance of datastores constructed using different data sources (DCLM-baseline and FineWeb-Edu) in Table R2. The results indicate that MassiveDS matches or even outperforms the datastores constructed with the latest high-quality pretraining data sources.

* Code: We uploaded our code to an anonymized repository and have sent the link to the AC.

 We will go into more detail about each of these in the relevant individual responses below.


**Table R1. Results of different base retrievers on 10% randomly sampled MassiveDS evaluated with Llama2-7B.**
| Name       | Retriever Type | Size  | Perplexity ↓ | Natural Questions ↑ | MMLU ↑ |
|------------|----------------|-------|--------------|----------------------|--------|
| Contriever-MSMACRO | dense          | 177M  | 4.221       | 33.2               | 49.2 |
| DRAGON [1]     | dense          | 110M  | 4.237       | 33.9           | 48.7 |
| GTR-Base [2]   | dense          | 110M  | 4.214   | 30.8               | 49.3 |

[1] Lin, Sheng-Chieh, et al. "How to train your dragon: Diverse augmentation towards generalizable dense retrieval."

[2] Ni, Jianmo, et al. "Large dual encoders are generalizable retrievers."

**Table R2. Results of different data sources for the datastore evaluated with Llama2-7B.**
| Source         | #Tokens (B) | NQ    | MMLU  |
|----------------|-------------|-------|-------|
| LM-only        | /           | 26.6  | 45.8  |
| DCLM-baseline [3]  | 100         | 31.4 | 49.0    |
| FineWeb-Edu [4]    | 100         | 30.4 | 49.4 |
| MassiveDS (ours)     | 100         | 33.2  | 48.8  |

[3] Li, Jeffrey, et al. "DataComp-LM: In search of the next generation of training sets for language models."

[4] Penedo, Guilherme, et al. "The FineWeb Datasets: Decanting the Web for the Finest Text Data at Scale."

**Figure R1. (See PDF)  Scaling performance on downstream tasks with different language models, corresponding to Figure 2 (c)-(f) with additional models included. The trends remain consistent.**

---

### Comment · Area_Chair_sjMa · 2024-08-12
**Reviewer-Author Discussions**

Dear reviewers: as you are aware, reviewer-author discussions phase is ending on Aug 13. We request you to kindly make use of the remaining time to contribute productively to these discussions. If you have not read and/or responded to author rebuttal, please do it asap so that the authors get a chance to respond to you. If you have more questions to ask or want further clarification from the authors, please feel free to do it.

Dear authors: please read the responses of the reviewers and if there are any pending questions/request-for-clarification, please attend to them.

---

### Decision · Program_Chairs · 2024-09-25

**Decision:**

Accept (poster)

**Comment:**

This work considers the problem of scaling data used at inference time in retrieval based LLMs. It presents MASSIVEDS, a
1.4 trillion-token datastore along with a pipeline to study efficiently the impact of different datastore features, such as data size, data filters, and decontamination strategies on language modeling as well as a few retrieval-based tasks that consume data retrieved from the datastore. It does a systematic study of the effect of scaling data on the performance and finds that scaling produces loglinear improvement without saturation.

Reviewers agree that scaling retrieval-based language models is an interesting problem, the paper is written well, experiments are systematic and demonstrate the value of scaling, code and data store can be potentially useful in future research in this field when open-sourced. Reviewers also found that some of the insights from the experiments are useful.

Reviewers expressed concern on the choice of RAG models used in the study, whether scaling would benefit more powerful RAG models, impact of scaling on efficiency of retrieval, lack of diversity in the tasks included in the scaling study, larger and more recent retrievers not considered, interplay between different types of retrievers and language models not studied, lack of comparative analysis of other open-source datastores.

I agree with the authors and the reviewers that the work will facilitate datastore scaling studies for retrieval-based tasks.